# A triple-defense electrocatalyst for robust seawater oxidation

Zixiao Li[1,2,8], Jie Liang[2,8], Shaohuan Hong[3,8], Yuchun Ren[1], Min Zhang[1], Shengjun Sun[1], Zhengwei Cai[1], Chaoxin Yang[1], Hefeng Wang[1], Yongsong Luo[4], Shanhu Liu[5], Yongchao Yao [6] ✉, Feng Gong [3] ✉, Xuping Sun [1,4] ✉ & Bo Tang [1,7] ✉

While coastal renewable energy-powered seawater electrolysis is highly promising for green $H_2$ production, the anodic chemical corrosion by aggressive chlorine chemistry and violent bubble release-induced physical damage to anodes are two long-standing issues that lead to inferior stability. Here we pursue integrating triple protection to a monolithic catalyst to concurrently alleviate chlorine chemistry and weaken external forces from bubble escaping/collapsing. The 1st and 2nd defenses are a Co-phosphate (Co-Pi) outer layer closely connected to CoP and well-dispersed nanosized $\gamma$-$MnO_2$ in/on Co-Pi, which collectively and preferentially filter out chloride ions approaching the catalytic sites based on their semipermeable natures. The 3rd defense comes from structural features that specialize in lessening the forces of bubble movements on the catalyst. A cage-shaped array composed of tip-connected nanowires with rough surfaces is verified to possess enhanced mechanical stability by theoretical simulations and experiments. This triple-protected electrocatalyst achieves a 3000-h electrolysis lifespan in real seawater during the ampere-level current density operation, demonstrating a multi-defense electrode design with guiding significance for wide applications.

A revolutionary energy upgrade is urgently needed as the conflict between the energy crisis and fast economic growth intensifies[1,2]. Hydrogen ($H_2$) is a vital industrial crude material and an up-and-coming fuel with clean combustion and a high calorific value, but traditional $H_2$ production technologies, using coals and natural gases, emit excess carbon dioxide ($CO_2$) and operate under high temperatures, making the development of sustainable and efficient low-carbon production technologies an on-going priority[3,4]. Renewable energy-powered water electrolysis has drawn a lot of research interest as an

opportunity to manufacture green $H_2$ while encouraging industrial sustainability[5–7]. Current mainstream water electrolyzers still require ultrapure water-based electrolyte feeds to enhance longevity, and the large-scale applications may exacerbate the shortage of freshwater resources[8,9]. Although the uneven distribution of freshwater resources poses regional challenges, advancements in seawater desalination largely mitigate freshwater supply constraints for water electrolysis. Nevertheless, the key advantage of seawater electrolysis lies not merely in conserving freshwater but in eliminating the need for

[1]College of Chemistry, Chemical Engineering and Materials Science, Shandong Normal University, Jinan, Shandong, China. [2]Institute of Fundamental and Frontier Sciences, University of Electronic Science and Technology of China, Chengdu, Sichuan, China. [3]MOE Key Laboratory of Energy Thermal Conversion and Control, School of Energy and Environment, Southeast University, Nanjing, Jiangsu, China. [4]Center for High Altitude Medicine, West China Hospital, Sichuan University, Chengdu, Sichuan, China. [5]College of Chemistry and Molecular Sciences, Henan University, Kaifeng, Henan, China. [6]Department of Laboratory Medicine, Precision Medicine Center, West China Hospital, Sichuan University, Chengdu, Sichuan, China. [7]Laoshan Laboratory, Qingdao, Shandong, China. [8]These authors contributed equally: Zixiao Li, Jie Liang, Shaohuan Hong. ✉e-mail: hatuu@wchscu.edu.cn; gongfeng@seu.edu.cn; xpsun@uestc.edu.cn; tangb@sdnu.edu.cn

desalination, thereby reducing system complexity and capital expenditure, particularly in coastal areas with limited freshwater access. In addition, natural seawater (serves as an inherent electrolyte) coupled with clean energy (e.g., offshore tidal, wind, and solar), offers great potential for promoting a low-carbon economy and the development of green hydrogen technologies[10–13]. However, the complexity of seawater, especially rich chloride ions (Cl⁻), makes the direct/indirect electrolysis demands a level of technological expertise that is far higher than that of classic and simple pure water-based electrolyzers[14,15]. Cl⁻ is so readily activated to more corrosive and noxious chlorine (Cl₂) via chlorine evolution reaction (CER) at anodic potentials ($E°_{anode} = 1.72$ V) and hypochlorous acid (HClO) via a disproportionation reaction[15], and the solution corrosivity soars as Cl₂ and HClO are produced. Since the potentials to activate Cl⁻ and hydroxide ion (OH⁻) differ by only a few hundred millivolts, even in seawater solution with elevated alkalinity, selective electrolysis with no CER is tough[15]. Moreover, higher current densities ($j$) create favorable potential ranges for CER to occur and a greater demand for surface ion (e.g., Cl⁻ and OH⁻) consumption. Thus, it is considerably harder to sustain CER-free seawater electro-oxidation processes at ampere-level $j$. In addition, catalyst structures are incessantly damaged by the bubbling of O₂ gas, which can drastically shorten the operating lifespan, especially when the $j$ increases[16,17]. Therefore, chlorine chemistry corrosion and the impact of gas movements together cause more rapid catalyst deactivation, and real effective strategies for such scientific challenges are highly desirable.

A wide variety of anode catalyst designs that focused on resisting chlorine chemistry corrosion, including creating a Cl⁻ repulsion barrier, optimizing electronic structures, customizing local environments, etc., are reported with better anti-corrosion performances[18–28], but the effectiveness of an individual catalyst design strategy can actually be limited. For instance, while creating anion-based barriers is acknowledged as a strategy promising for preventing the diffusion and migration of Cl⁻ to catalyst surfaces[18,19,29], trace Cl⁻ can still diffuse and migrate to catalytic sites to be oxidized, and thus such protection does not completely block CER. More specialized surface designs toward comprehensive inhibition of CER are highly desired to make the catalyst's lifespan longer. Note that catalytic activity degradation comes not only from chlorine species-induced corrosion but also from the continuous physical attack on the catalyst by highly aggressive bubble movements (e.g., coalescence) at high $j$[16,17]. Apart from chemical stability, seawater electrolysis catalysts' mechanical robustness at high $j$ is overlooked. Irregular or irrational micro-assembly of nanostructures will further hinder the flow of bubbles and subject catalysts to greater local stresses, which is highly detrimental to achieving stable electrolysis[16,17,30,31].

In this article, we report a monolithic array electrode with a nanoscale assembly that integrates triple protection to realize stable electrolysis in alkaline seawater. This elegant three-in-one design includes a negatively charged phosphate species-enriched passivating outer surface as the 1st defense to preferentially repel Cl⁻, highly dispersed nanosized γ-MnO₂ as the 2nd defense to selectively and more thoroughly filter out Cl⁻, and a special tip-connected nanowire array as the 3rd defense architecture to minimize the force of gas bubbles on the 3D arrayed catalyst. Each part contributing to the longer electrolysis lifespans in salt-rich seawater is thoroughly discussed with strong experimental support and theoretical calculations/simulations. Under multiple protective shields, the self-protecting electrode enables state-of-the-art performance towards electrochemical alkaline seawater oxidation (eASO), maintaining high-activity electrolysis for at least 3000 h under a fixed ampere-level $j$. This work is different from any previous seawater catalyst designs, which either solely focused on improving mechanical strength or merely targeted at enhancing chemical stability, because it simultaneously boosts the mechanical and chemical lifespans of an electrode and realizes the integration of ingenious protection strategies.

## Results

In general, the lifetime of an electrode in seawater electrolytes depends not only on chemical stability but also on mechanical stability. Chlorine chemistry-induced catalyst depletion and metal substrate depletion badly weaken chemical stability by leaching and dissolution, explaining why a lot of research focuses on finding the most effective way to lessen the amount of Cl⁻ that reaches catalysts and even cross/penetrates into the inner substrate. As shown in Fig. 1a, an efficient method for decreasing the content of surface Cl⁻ and making it harder to approach the catalyst is to create a barrier of protection that repels the Cl⁻[10,11]. While inhibiting chloride corrosion to some extent, these strategies do not mitigate the ongoing physical damage to the catalyst caused by the violent gas bubble motion under ampere-level $j$. This kind of damage to an electrode also increases with increasing $j$, eventually leading to catalyst fragmentation and detachment. While the electrode mechanical stability can increase by preparing mechanically tough catalysts or enhancing the catalyst-substrate adhesion, the continuous attack of the bubbles as the root of the problem remains unaddressed. Array-like nanostructures, due to the empty space within them, can reduce the shock of external forces to a certain extent. This empty space, i.e., the 3D channels within the arrayed catalyst, can provide good microdiffusion and interfacial diffusion to better prevent oxygen evolution reaction (OER)-generated O₂ accumulation in the electrode (Fig. 1b)[17]. In other words, a well-regulated nanoarray enables better bubble/liquid flows to exhibit better mechanical stability by alleviating the local tensile and vibrational forces induced by the escape/collapse of O₂ bubble. A great deal of research into optimizing bubble release on OER electrodes, however, was focused solely on pure water-based electrolysis[17]. The challenge is to combine designs that simultaneously improve mechanical and chemical stability on the same electrode at the nanoscale (Fig. 1c). We demonstrate a triple-protected array electrode that is greatly optimized in terms of chemical composition as well as architectural design. The electrode possesses not only outer surface amorphous Co-phosphate (Co-Pi) as the 1st defense and uniformly distributed γ-MnO₂ nanoparticles as the 2nd defense to jointly and selectively filter out Cl⁻, but also a tip-connected cage-like array structure as the 3rd defense that facilitates gas release and reduces local stress. The effectiveness of each defense and how it works are explored and discussed in detail in this work. Supplementary Table 1, which lists and illustrates a number of the research conducted in recent years on improving the OER electrode lifespan, highlights the significance of concurrently solving chlorine chemistry corrosion-induced instability and physical damage-induced mechanical instability.

### Electrode with a triple defense design

The target electrode was synthesized using a self-supported CoP nanowire catalyst as a precursor by simple immersion within a KMnO₄ solution. The surface of CoP oxidizes into other Co and P species due to surface MnO₄⁻ adsorption, and MnO₂ forms on the solid-liquid interface. Such spontaneous redox reactions between MnO₄⁻ and CoP not only create surface well-dispersed MnO₂ and Co-Pi in situ to filter out Cl⁻ as strong selective dual barriers but also change the arrangement and surface roughness of nanowires to constitute a more specialized cage-like array structure (Fig. 2a). The resulting electrode with Ni foam supported CoP (CoP/NF) as the core and surface Co-Pi and MnO₂ can be denoted as MnO₂@Co-Pi@CoP/NF. For comparison, Co-Pi coated CoP on NF (Co-Pi@CoP/NF) was also synthesized by an electrochemical anodization treatment of CoP/NF. As shown in Supplementary Fig. 1, X-ray diffraction (XRD) pattern of CoP/NF shows peaks for CoP (PDF#29-0497) and the Ni substrate. XRD pattern of MnO₂@Co-Pi@CoP/NF still shows peaks of CoP. Also, peaks at 22.4°, 37.1°, 42.5°, and 56.3° that belong to γ-MnO₂ (PDF#14-0644) are also present, thus indicating that soaking CoP/NF in the KMnO₄ solution can create crystalline MnO₂ and amorphous Pi in situ via the

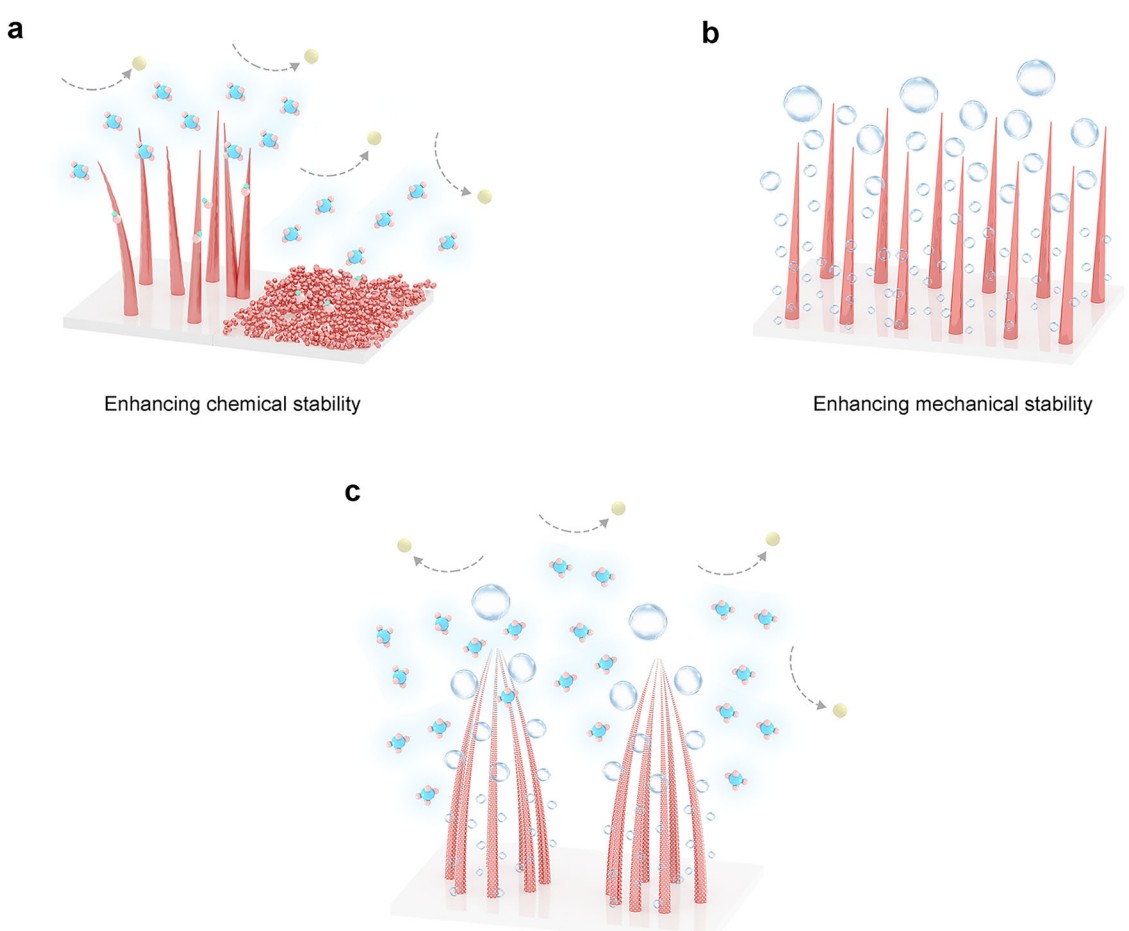

**Fig. 1 | Enhancing chemical and mechanical stability of electrode at the same time. a** Electrode designs for chemical stability enhancement only, whether self-supported or binder-based electrodes. **b** Electrode designs for mechanical stability enhancement only. **c** Next-generation electrode designs for simultaneously improving chemical and mechanical stability.

spontaneous redox process. Besides, the diffraction peaks of CoP, with little shift but weakened intensities, are attributed to surface oxidation of the phosphides to phosphates. As for Co-Pi@CoP/NF, the anodic polarization treatment also reduces the crystallinity of CoP and generates amorphous/low-crystallinity phosphates. Furthermore, CoP/NF has no distinct Raman spectral bands over a wide range of 700 to 1250 cm⁻¹, but Co-Pi@CoP/NF and MnO₂@Co-Pi@CoP/NF both show Raman bands at 984.9 and 1060.5 cm⁻¹ belonging to phosphorus species like $HPO_4^{2-}$ and $PO_4^{3-}$ (Supplementary Fig. 2)[32–34], again indicating the successful introduction of phosphate-related species. Also, only MnO₂@Co-Pi@CoP/NF has a distinguishable Raman band at 651.5 cm⁻¹ (Mn-O), which is attributed to γ-MnO₂[35]. Scanning electron microscopy (SEM) images of CoP/NF (Fig. 2b) and MnO₂@Co-Pi@CoP/NF (Fig. 2c) reveal two rather different morphological architectures composed of nanowires with similar sizes. Self-supported CoP on NF exhibits a random arrangement of nanowires with little regularity in the nanowire head/tip direction. Such nanowires are bundled up to form one cluster after another, and the cluster-to-cluster arrangement is disordered as well, with large crossover and overlapping areas. Co-Pi@CoP on NF likewise exhibits a comparable disordered nanoneedle arrangement as CoP on NF, but the outer surface roughness grows by the anodic treatment that produces Co-Pi (Supplementary Fig. 3). As for MnO₂@Co-Pi@CoP/NF, soaking-induced oxidation etching engineering changes the arrangement of nanowires into integration of multiple 3D nanocage-like structures composed of head/tip-connected nanowires. Compared to CoP/NF and Co-Pi@CoP/NF,

MnO₂@Co-Pi@CoP on NF demonstrates a far rougher surface, most likely as a result of the co-presence of surface phosphates and MnO₂. Transmission electron microscopy (TEM) image of MnO₂@Co-Pi@CoP (Fig. 2d) further illustrates the nanowire tip/head-joining, nanocage-shaped, and roughened surface structural features, as characterized by SEM. High-resolution TEM (HRTEM) images (Fig. 2e) identify the internal CoP core, amorphous Co-Pi, and dispersed MnO₂ nanoparticles. In addition, HRTEM images from various regions indicate that the average thickness of the amorphous Co-Pi layer is approximately 9 nm (Supplementary Fig. 4). The good crystallinity of both MnO₂ and CoP are supported by a lattice spacing of 0.217 nm that corresponds to the (300) plane of MnO₂ as well as a spacing of 0.172 nm that matches with the (103) plane of CoP. Energy-dispersive X-ray spectroscopy data (Fig. 2f) can verify the uniform distribution of Co, P, O, and Mn elements throughout the one-dimensional MnO₂@Co-Pi@CoP.

X-ray photoelectron spectroscopy (XPS) was carried out to explore valence changes and charge transfers for MnO₂@Co-Pi@CoP. High-resolution Co 2$p$ XPS spectra of CoP, Co-Pi@CoP, and MnO₂@Co-Pi@CoP (Supplementary Fig. 5a) are mainly split into Co 2$p_{3/2}$ and Co 2$p_{1/2}$. As for CoP, peaks of Co$^{δ+}$ 2$p_{3/2}$ and Co$^{δ+}$ 2$p_{1/2}$ appear at 777.6 and 792.5 eV, corresponding to the Co–P bond[36], and peaks at 780.8 and 797.1 eV are related to Co$^{2+}$ 2$p_{3/2}$ and Co$^{2+}$ 2$p_{1/2}$ from surface oxidized species. Higher occupation of oxidized Co and less Co$^{δ+}$ are found in the spectra for Co-Pi@CoP and MnO₂@Co-Pi@CoP due to surface species with higher valences. MnO₂@Co-Pi@CoP shows a

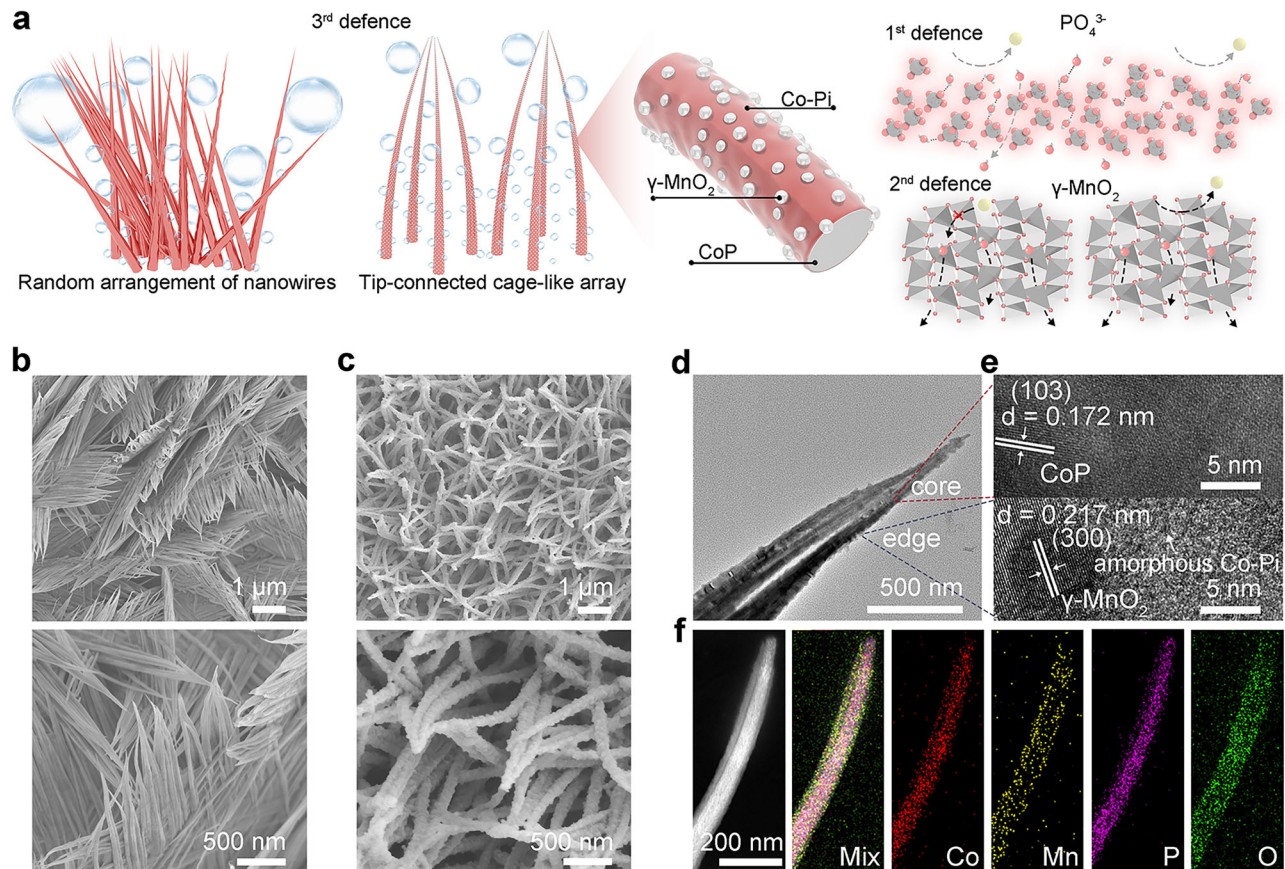

**Fig. 2 | Material characterizations. a** Brief schematic diagram of the triple defense. **b** SEM images of CoP/NF showing a random arrangement of nanowires. **c** SEM images of MnO$_2$@Co-Pi@CoP/NF showing the tip-connected nanocage-like array. **d** TEM and **e** HRTEM images of MnO$_2$@Co-Pi@CoP. **f** Scanning TEM and the mapping images.

lower Co–O content than that of Co-Pi@CoP, probably because of a lower oxidation degree and the decorated MnO$_2$ occupying the sites of Co-Pi. The P 2$p$ XPS spectrum of CoP is divided into P–O (133.6 eV) and P–metal (P–M, 128.9 eV) peaks, and the P–M peak can be divided into P 2$p_{3/2}$ (128.8 eV) and P 2$p_{1/2}$ (129.7 eV) (Supplementary Fig. 5b). A same kind of species distribution (i.e., P–O and P–M) is found in Co-Pi@CoP and MnO$_2$@Co-Pi@CoP, but the trend of P–M content is as follows: CoP > Co-Pi@CoP > MnO$_2$@Co-Pi@CoP. Therefore, the surface phosphide amounts on the three samples decrease accordingly, which are consistent with larger Co–O sub-peaks and smaller P–M sub-peaks of Co-Pi@CoP and MnO$_2$@Co-Pi@CoP. XPS spectra in Mn 3$s$ and Mn 2$p$ regions for MnO$_2$@Co-Pi@CoP confirm the successful fabrication of MnO$_2$ by a spontaneous redox reaction between CoP and MnO$_4^-$ (Supplementary Fig. 5c,d)[37–39].

Based on the material characterizations, a short overview of the integration of triple defense is briefly described. The special features of the MnO$_2$@Co-Pi@CoP/NF (Fig. 2a) are the following aspects. First of all, a conductive CoP core can provide a high-speed way for electron transfers, and amorphous Co-Pi derived from the CoP can act as the 1$^{st}$ defense to selectively repel Cl$^-$. The long-range disordered Co-Pi layer has no grains (i.e., less ion channels) for Cl$^-$ to pass/cross, thus preventing the Cl$^-$ in seawater from corroding the underlying Ni substrate[40]. More functions of the PO$_4^{3-}$-based barrier will be mentioned later. Secondly, the MnO$_2$ nanoparticles dispersed in the Co-Pi layer act as the 2nd defense can also filter out Cl$^-$ and cooperate with the PO$_4^{3-}$-based layer to more thoroughly prevent the occurrence of CER. The cage-like architectures composed of tip-connected nanowires can guide bubble detachment/movement, achieve higher flow velocities around the array, and reduce the forces on the array from bubbles. For instance, gas bubbles may move along curved pillars (i.e.,

nanowires) to the cage-like architecture's top part for a rapid detachment. Also, the multidimensional features consisting of 1D Co-Pi@CoP and 0D MnO$_2$ can further facilitate the bubble release. The rougher nanowire surface due to the MnO$_2$ nanoparticles decorated at the outer layers can prevent bubble adhesion better and facilitate bubble migration on the nanoarray. In consequence, the tip-connected nanoarray will be subjected to considerably more uniform tensile and vibrational forces from the bubble movements and less localized stresses. Such physical defense would thus minimize the unwanted external forces and stabilize the arrayed catalyst.

**Catalytic activity improvements**

The overpotential ($\eta$) required by an electrode to attain 1000 mA cm$^{-2}$ ($\eta_{1000}$) can act as an indicator of whether the electrode is suitable for industrial-scale electrolysis systems. As shown in Supplementary Fig. 6a, polarization curves (85%-iR correction) are recorded within a potential window from 1.0 to 1.75 V vs. the reversible hydrogen electrode (RHE), and MnO$_2$@Co-Pi@CoP/NF attains 1 A cm$^{-2}$ with the smallest $\eta_{1000}$ of 352 mV compared with Co-Pi@CoP/NF (402.6 mV) and CoP/NF (440 mV). To better evaluate the intrinsic activity of the catalysts and minimize the interference from gas bubble release, local temperature, and pH fluctuations under high $j$, we also compare the $\eta$ at a low $j$ of 10 mA cm$^{-2}$, which are 266, 247, and 228 mV for CoP/NF, Co-Pi@CoP/NF, and MnO$_2$@Co-Pi@CoP/NF (Supplementary Fig. 6b), respectively. The uncorrected polarization curves further confirm the favorable catalytic activity of the MnO$_2$@Co-Pi@CoP/NF compared to the counterpart samples (Supplementary Fig. 6c). Importantly, the $\eta_{1000}$ for MnO$_2$@Co-Pi@CoP/NF is competitive with those of many recent anodes (Supplementary Table 2). As shown in Supplementary Fig. 7, the Tafel slope values and corresponding log|$j$| data for CoP/NF,

Co-Pi@CoP/NF, and MnO$_2$@Co-Pi@CoP/NF indicate that the kinetic advantage of MnO$_2$@Co-Pi@CoP/NF becomes more pronounced at higher $j$. While both MnO$_2$ and Co-Pi are introduced to enhance stability, the experiments demonstrate that they also improve the activity rather than impair it. Potential-dependent Nyquist plots and the corresponding Bode plots are given in Supplementary Figs. 8, 9 to further explore the electron transfer and reaction kinetics. MnO$_2$@Co-Pi@CoP/NF exhibits the lowest charge transfer resistances under different potentials. Also, transition peaks in Bode plots change differently in a low-frequency region (0.01 to 10 Hz) for the three electrodes, and the phase angles of MnO$_2$@Co-Pi@CoP/NF decrease the fastest, signifying that rapid OER kinetics and fast intermediate deprotonation[41,42]. Intrinsic catalytic activities can be quantified with cyclic voltammetry (CV) measurements (Supplementary Fig. 10a–c), and the turnover frequency (TOF) value of MnO$_2$@Co-Pi@CoP/NF (0.28 s$^{-1}$) at 0.35 V is twice that of Co-Pi@CoP/NF (0.14 s$^{-1}$) and is 3.5 times that of CoP/NF (0.08 s$^{-1}$), which suggesting the highest intrinsic activity towards the eASO (Supplementary Fig. 10d). Polarization curves measured at different temperatures (Supplementary Fig. 11a–c) were used to obtain the activation energy of the catalytic material based on the Arrhenius equation. The calculated activation energy of MnO$_2$@Co-Pi@CoP/NF is 4.07 kJ mol$^{-1}$, much lower than 11.1 kJ mol$^{-1}$ of CoP/NF and 11.01 kJ mol$^{-1}$ of Co-Pi@CoP/NF (Supplementary Fig. 11d), indicating that the MnO$_2$@Co-Pi@CoP/NF requires a lower energy barrier to achieve OER.

Fourier-transformed alternating current voltammetry (FTacV) provides powerful technical support for probing intrinsic active sites and conducting detailed kinetic reaction analysis, while effectively mitigating interference from non-Faradaic processes[43–45]. The current distribution in the potential window can be divided into three regions, representing the oxidation of Co species, structural transformation to CoOOH, and eASO (Supplementary Fig. 12a). The corresponding integral areas of three regions (Supplementary Fig. 12b) show more visually the relative magnitude of the current values in each region for the three materials. In region 1, CoP/NF shows the highest current response due to its larger amounts of low-valence Co$^{\delta+}$ species being oxidized compared with the other two anodes. MnO$_2$@Co-Pi@CoP/NF exhibits a higher current response than that of Co-Pi@CoP/NF, probably resulting from more low-valence Co species. The current response in region 2 of MnO$_2$@Co-Pi@CoP/NF is the highest, indicating the fastest and easiest transformation to the active species (i.e., CoOOH). Similarly, a comparably high current response of Co-Pi@CoP/NF indicates that high-valence Co species of Co-Pi would contribute to the structural transformation. With more active CoOOH, MnO$_2$@Co-Pi@CoP/NF obtains the highest current response in region 3, representing its highest eASO activity, which is consistent with the results of linear sweep voltammetry (LSV) data. In situ Raman spectra of CoP/NF, Co-Pi@CoP/NF, and MnO$_2$@Co-Pi@CoP/NF again verify that MnO$_2$@Co-Pi@CoP requires the smallest $\eta$ to generate active CoOOH (Supplementary Fig. 13).

Under alkaline conditions, the OER involves multiple proton/charge-transfer steps, typically deprotonation of OH$_{(ads)}$ or OOH$_{(ads)}$ to O–O intermediates, which are tightly coupled with proton and electron transfer processes[46]. Polarization curves recorded with solution pH ranging from 14 to 12.5 are shown in Supplementary Fig. 14. The $\rho^{RHE}$ ($\partial\log(j)/\partial$pH) values are 0.545, 0.924, and 0.936 for CoP/NF, Co-Pi@CoP/NF, and MnO$_2$@Co-Pi@CoP/NF, thus implying that the presence of phosphate makes the catalytic activity more pH dependent[19]. Co-Pi@CoP/NF and MnO$_2$@Co-Pi@CoP/NF both show much smaller $j$ and lower Tafel slope values by replacing KOH with tetramethylammonium hydroxide (TMAOH) in alkaline seawater, whereas the activity of CoP changes little (Supplementary Fig. 15a,b). Raman spectra also reveal the peaks of TMA$^+$ for Co-Pi@CoP/NF and MnO$_2$@Co-Pi@CoP/NF after being activated in 1 M TMAOH + seawater (Supplementary Fig. 15c). Therefore, the phosphates and nanosized

MnO$_2$ should activate lattice oxygen redox reactions during eASO via non-concerted proton-electron transfers. Online differential mass spectrometry can detect O$_2$ gas generated from the electrodes to directly clarify the reaction pathways. CoP/NF, Co-Pi@CoP/NF, and MnO$_2$@Co-Pi@CoP/NF were activated in 0.1 M KOH solution containing H$_2^{18}$O by CV scans. Then, labeled electrodes operated in 0.1 M KOH solution, and the gas was monitored by online mass spectrometry. The signal for $^{16}$O$^{16}$O is detected for CoP/NF, whereas an additional signal for $^{16}$O$^{18}$O is detected for Co-Pi@CoP/NF and MnO$_2$@Co-Pi@CoP/NF (Supplementary Fig. 16). The isotope labeling results again confirm that CoP/NF follows an adsorbate evolution mechanism during OER, while Co-Pi@CoP/NF with phosphates and MnO$_2$@Co-Pi@CoP/NF with phosphates and MnO$_2$ generate O$_2$ more through the lattice oxygen oxidation pathway.

## Improved lifespans under a triple protection

Chronopotentiometry curves of CoP/NF, Co-Pi@CoP/NF, and MnO$_2$@Co-Pi@CoP/NF recorded in alkaline seawater at the $j$ of up to 2 A cm$^{-2}$ can reflect the defense-based improvements in electrolysis lifespans (Fig. 3a). As expected, MnO$_2$@Co-Pi@CoP/NF easily achieves the longest operation time of 800 h, more than 5 h of CoP/NF, 117 h of Co-Pi@CoP/NF, and 350 h of MnO$_2$@CoO$_x$/NF (Supplementary Fig. 17). A piece of Co-Pi@CoP/NF and a piece of MnO$_2$@Co-Pi@CoP/NF of the same size are immersed in natural seawater to visualize the corrosion resistance of the electrodes (inset of Fig. 3a). 24-h soaking severely corrodes the Co-Pi@CoP/NF, and a lot of fragments appear at the bottom of the vial, which suggests the limited chlorine-corrosion resistance. Note that the MnO$_2$@Co-Pi@CoP/NF retains its original shape well without any fragments in the vial after 24-h soaking. The high electrode integrity demonstrates that the dual protection of MnO$_2$ and Co-Pi indeed inhibits chlorine chemical corrosion. Notably, MnO$_2$@Co-Pi@CoP/NF continuously catalyzes eASO for at least 3000 h at 1 A cm$^{-2}$ without notable activity degradation (Fig. 3b), thus establishing high eASO stability. XPS spectra in the Cl 2$p$ region for the corresponding electrodes after stability tests at 2 A cm$^{-2}$ (Fig. 3c) show the strongest peak intensity for CoP, a slightly weakened peak intensity for Co-Pi@CoP, and the weakest peak intensity of MnO$_2$@Co-Pi@CoP, which indicates that surface chlorine species decrease with the increase in protection degree. The Cl 2$p$ spectra are further deconvoluted by different components, with the Co−Cl peaks at the smallest binding energy[47,48]. The surface XPS data of MnO$_2$@Co-Pi@CoP shows minimal Co−Cl bonding. Moreover, almost no chlorine species can be found on MnO$_2$@Co-Pi@CoP/NF after the 3000-h durability test at 1 A cm$^{-2}$ (see XPS data at the bottom of Fig. 3c). Since the phosphorus content in Co-Pi@CoP is comparable to that in MnO$_2$@Co-Pi@CoP, the CER-inhibition ability based on PO$_4^{3-}$ alone is verified to be feasible but limited. Despite this, in terms of the multiplicative increase in the lifetime (i.e., a 23.4-fold increase), this oxyanion-based strategy is still effective. Moreover, active chlorine contents in the seawater electrolytes after long-term eASO were detected using the N,N-diethyl-p-phenylenediamine-based method (Supplementary Fig. 18). We have detected active chlorine produced (hypochlorite, ClO$^-$) at the same electrolysis conditions (operation time and $j$) to give a direct comparison of selectivity for catalysts. As shown in Supplementary Fig. 19, UV-visible-near-infrared spectra of ClO$^-$ generated from CoP/NF, Co-Pi@CoP/NF, and MnO$_2$@Co-Pi@CoP/NF after 50-h and 100-h electrolysis at 1 A cm$^{-2}$ reveal a clear trend: MnO$_2$@Co-Pi@CoP/NF produced the least amount of ClO$^-$, followed by Co-Pi@CoP/NF, while CoP/NF produced the most, confirming that MnO$_2$@Co-Pi@CoP/NF exhibits the improved selectivity. As for lifespan tests, the content of ClO$^-$ detected in the solution corresponding to MnO$_2$@Co-Pi@CoP/NF is only 26% of the content for Co-Pi@CoP/NF and 16.5% for CoP/NF (Supplementary Fig. 20). The ClO$^-$ content produced by MnO$_2$@Co-Pi@CoP/NF after 1500-h durability test at 1 A cm$^{-2}$ is only 8% of that produced by CoP/NF after only 5 h of electrolysis at 2 A cm$^{-2}$

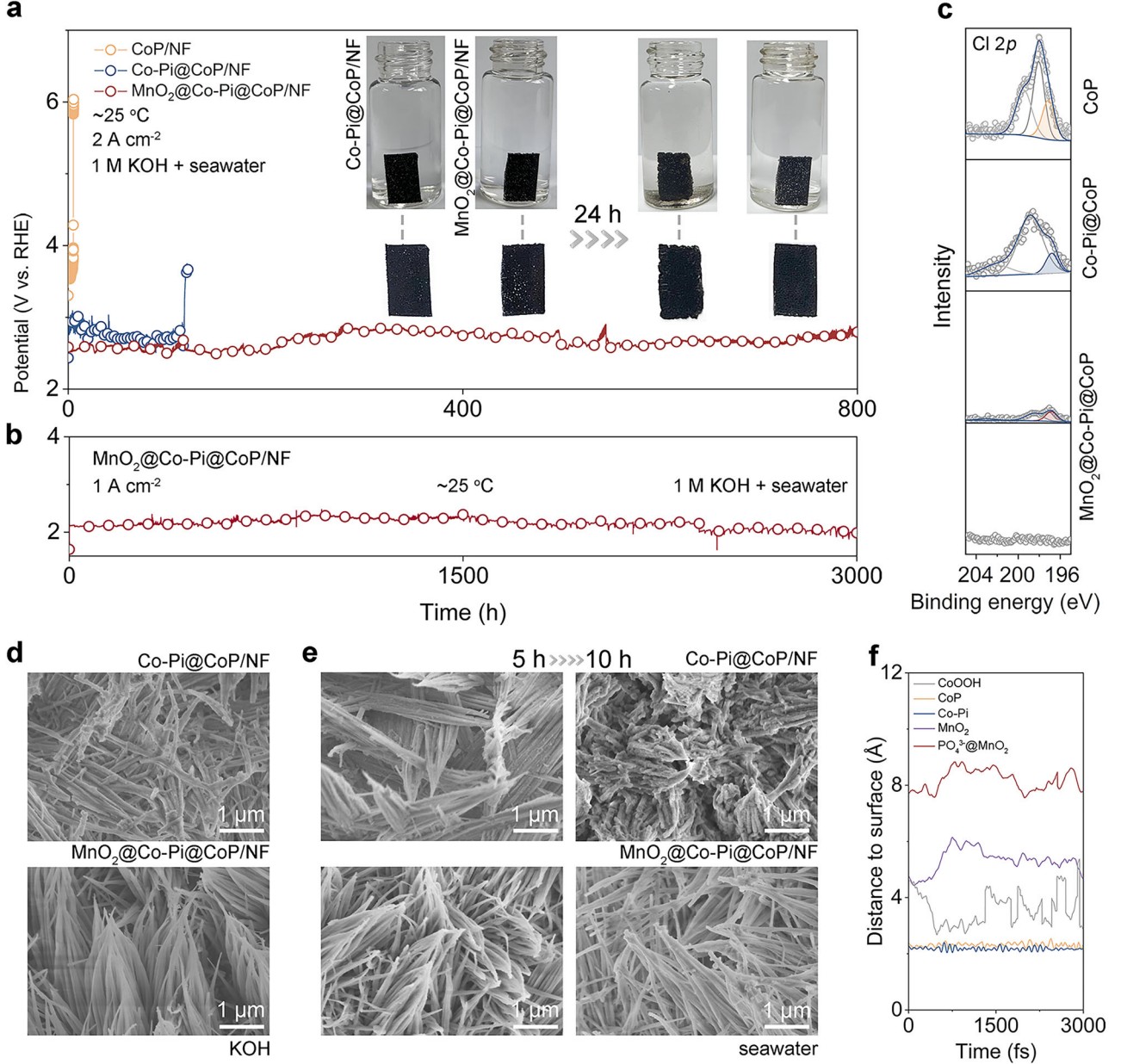

**Fig. 3 | Improvements in the lifespan of MnO₂@Co-Pi@CoP/NF with a triple defense. a** Long-term electrolysis durability comparison for the different anodes under 2 A cm⁻² without *iR* correction. The inset photos show MnO₂@Co-Pi@CoP/NF and Co-Pi@CoP/NF before and after 24 h soaking in natural seawater. **b** 3000 h of electrolysis under 1 A cm⁻² without *iR* correction. **c** XPS data for the related anodes in Cl 2*p* region. **d** SEM images after 10 h OER tests. **e** SEM images after eASO tests under high *j*. **f** AIMD data.

(Supplementary Fig. 21). Moreover, MnO₂@Co-Pi@CoP/NF shows a nearly 100% O₂ Faradaic efficiency via the drainage method at 500 mA cm⁻² (Supplementary Fig. 22). In addition, MnO₂@Co-Pi@CoP/NF exhibits the highest anti-corrosion potential of 1.037 V vs. RHE compared to CoP/NF (0.871 V vs. RHE) and Co-Pi@CoP/NF (0.932 V vs. RHE) in alkaline seawater (Supplementary Fig. 23). Since methanol molecules are able to nucleophilically attack electrophilic *OH, the polarization curves of CoP/NF, Co-Pi@CoP/NF, and MnO₂@Co-Pi@CoP/NF were recorded in alkaline seawater with and without 0.5 M methanol. As shown in Supplementary Fig. 24, the methanol oxidation-induced differences in *j* are calculated from the filled area. CoP/NF has the smallest differences in *j*, whereas Co-Pi@CoP/NF has a larger difference in *j*, and MnO₂@Co-Pi@CoP/NF has the largest difference in *j*, showing that MnO₂@Co-Pi@CoP/NF possesses the highest adsorption abilities of *OH. Therefore, the PO₄³⁻-

based protective barrier can alleviate chlorine chemistry to a certain extent, and the presence of surface nanosized MnO₂ and the special tip-connected morphology should play vital roles in further enhancing the electrode stability. Furthermore, an anion exchange membrane seawater electrolyzer assembled directly using a piece of MnO₂@Co-Pi@CoP/NF as the binder-free anode also shows good performance (Supplementary Fig. 25).

The post-OER morphologies of MnO₂@Co-Pi@CoP/NF can be compared to those of the counterpart anode (i.e., Co-Pi@CoP/NF) to directly verify the improved mechanical property. As shown in Fig. 3d, MnO₂@Co-Pi@CoP/NF shows good structural integrity after continuous electrolysis at 0.5 A cm⁻² for 10 h in KOH solution, while the nanowire morphology of Co-Pi@CoP/NF is seriously damaged, with unfavorable features such as uneven sizes of nanowires, broken nanowires, and agglomeration. After establishing the baseline of

enhanced mechanical stability, the morphologies of the anodes after eASO at high reaction $j$ were also compared. It is evident that the counterpart anode suffers from corrosion and bubble attack during the 5-h electrolysis operation, yet there is no change in the nanowire morphology of MnO$_2$@Co-Pi@CoP on NF (Fig. 3e). A longer electrolysis operation leads to more severe corrosion and agglomeration of nanowires on the counterpart anode, with apparent catalyst detachment (Fig. 3e). In contrast, the original morphologies with tip-connected nanowire structures are still retained for MnO$_2$@Co-Pi@CoP on NF. It is thus considered that both the mechanical and chemical stability are improved.

Co-Pi@CoP/NF already shows improved corrosion resistance and stability due to the protective outer layer of phosphates, and MnO$_2$@Co-Pi@CoP/NF even demonstrates a higher level of operational stability, thereby reflecting the effectiveness of introducing surface MnO$_2$ in achieving robust stability. Considering that, ab initio molecular dynamics (AIMD) simulations were conducted to unravel behaviors of Cl$^-$ on different catalytic surfaces. The explicit solvent model consisting of 45 H$_2$O, one K$^+$, one OH$^-$, and one Cl$^-$ is used for simulations (Supplementary Fig. 26), and each system is fully equilibrated to construct the electrochemical interface close to reaction conditions (see "Methods"). As depicted in Supplementary Fig. 26, γ-MnO$_2$ (100) and PO$_4^{3-}$@γ-MnO$_2$ (100) both turn out to have higher coverage of oxygen-containing intermediates, including *O, *OH, and *H$_2$O, while fewer oxygen-containing intermediates are generated on CoOOH (001), CoP (011), and Co-Pi (001), thus indicating γ-MnO$_2$ (100) and PO$_4^{3-}$@γ-MnO$_2$ (100) possess strong adsorption of these species. As a result of different coverages of the oxygen-containing intermediates, PO$_4^{3-}$@γ-MnO$_2$ (100) displays significant resistance to Cl$^-$ than other materials. As shown in Fig. 3f, the PO$_4^{3-}$@γ-MnO$_2$ (100) has the longest average Cl$^-$-to-surface distance of 8.16 Å (2.32 Å for CoP (011), 2.18 Å for Co-Pi (001), 3.59 Å for CoOOH (001), and 5.48 Å for γ-MnO$_2$ (100), respectively), demonstrating that the PO$_4^{3-}$@γ-MnO$_2$ (100) has the strongest resistance to Cl$^-$. The surface of CoOOH (001) is defined by the O atoms in the top layer, and the Cl$^-$ is captured by the Co atom, which is reconstructed by *OH species to acquire a higher position than the defined surface (Supplementary Fig. 26). Therefore, the average Cl$^-$-to-surface distance is much longer than the Co−Cl bond. It should be noted that Cl$^-$ anions are adsorbed by the active sites of all the materials except γ-MnO$_2$ (100) and PO$_4^{3-}$@γ-MnO$_2$ (100) (Supplementary Fig. 26). In addition, although *O species exist on γ-MnO$_2$ (100) and PO$_4^{3-}$@γ-MnO$_2$ (100), no *OCl species can be found during the AIMD process, further demonstrating the considerable resistance of γ-MnO$_2$ to Cl$^-$.

As shown in Supplementary Table 3, MnO$_2$@Co-Pi@CoP/NF achieves a leading stability performance. All the studies selected are state-of-the-art anodes reported within the past year, of which only CoFe-Ci@GQD/NF was tested in simulated alkaline seawater (i.e., a solution with KOH + NaCl). MnO$_2$@Co-Pi@CoP/NF thus holds an absolute advantage in terms of the electrolysis stability in real seawater electrolytes. Meanwhile, the decay voltage ($D_V$) of the reported catalysts is calculated using the criterion $D_V = (V_1 - V_2)/t$, where $V_1$ and $V_2$ represent average voltages of the first and last 10% eASO time. However, the calculated $V_1$ and $V_2$ of MnO$_2$@Co-Pi@CoP/NF at 1 A cm$^{-2}$ are 2.13 and 2.04 V vs. RHE, respectively, which means that its $D_V$ cannot be calculated but indicates its promising stability for 3000-h electrolysis. Importantly, MnO$_2$@Co-Pi@CoP/NF is prepared without the use of any noble metals. The robust stability at ampere-level $j$, along with the relatively low cost of the materials, highlights a possible entry point for reducing the costs of seawater electrolyzers while providing long-term operation with high activity.

## 1st defense: PO$_4^{3-}$-enriched barriers

The introduction of inorganic anions, such as CO$_3^{2-}$, SO$_4^{2-}$, and PO$_4^{3-}$, to the catalyst surface presents a real effective strategy toward interfering with unwanted chlorine chemistry[18,19,29,49–51]. A volcano plot

based on the charge number ($Z$) and radius ($r$) showed that PO$_4^{3-}$ has a satisfying ionic potential as well as $Z \times r$ value, thus disfavoring the proximity of Cl$^-$[52]. The use of CoP as a core for MnO$_2$@Co-Pi@CoP actually fulfills several key purposes. First of all, transition metal phosphides, especially 1D CoP here, possess higher electronic conductivity than the corresponding oxo/hydroxides to benefit electrocatalysis with improved charge transfer[52]. Secondly, before electrolysis, the CoP acts as a phosphorus source, allowing the in situ generation of cation-selective PO$_4^{3-}$-rich passivating barriers (i.e., Co-Pi) in close contact with the inner CoP as the 1st defense to repel Cl$^-$. During electrolysis, H$_2$O at the interface and PO$_4^{3-}$ adsorbed on the catalyst naturally interact to form a semipermeable skeleton via hydrogen bonding effects, which selectively repels Cl$^-$ without markedly impeding the OH$^-$ diffusion because Cl$^-$ is a weak hydrogen bond receptor[53]. Even if the catalyst surface part is reconstructed to (oxy) hydroxides, a strong hydrogen bonding between OH$^-$ and the (oxy) hydroxides (e.g., CoOOH) can keep OH$^-$ from being electrostatically repelled[54]. In addition, the amorphous Co-Pi should have fewer ion channels for Cl$^-$ to cross[40], thus better protecting the internal electron-conducting NF and CoP to minimize the etching by Cl$^-$, which is another basis for stable electrolysis. Last but not least, the well-known and reliable pH buffering nature of PO$_4^{3-}$ can weaken the local pH drop to a certain extent at high $j$, further safeguarding the chemical stability of the MnO$_2$@Co-Pi@CoP/NF.

Surface phosphorus species in MnO$_2$@Co-Pi@CoP/NF before, during, and after tests were analyzed carefully to reveal the effects and the rationale for the 1st line of defense. Time-of-flight secondary ion mass spectrometry (TOF-SIMS) was adopted to probe the concentrations and distribution of key anions (PO$_4^{3-}$, OH$^-$, and Cl$^-$) of selected areas on the electrodes after 10 h of eASO under high $j$ (Fig. 4a). The trend of Cl$^-$ concentration that can be directly mapped by the relative intensity of Fig. 4a is as follows: CoP/NF > Co-Pi@CoP/NF > MnO$_2$@Co-Pi@CoP/NF, whereas the concentration of OH$^-$ for CoP/NF is less than that for Co-Pi@CoP/NF or MnO$_2$@Co-Pi@CoP/NF (i.e., declining Cl$^-$ and rising OH$^-$ signal intensities for both the Co-Pi@CoP/NF and MnO$_2$@Co-Pi@CoP/NF). Concurrently, the PO$_4^{3-}$ concentration on the CoP surface is substantially less than the PO$_4^{3-}$ concentration on Co-Pi@CoP or MnO$_2$@Co-Pi@CoP. TOF-SIMS mapping thus reveals three key information: (1) CoP/NF exhibits the lowest Cl$^-$ repulsion ability and suffers from more surface Cl$^-$ absorption; (2) the PO$_4^{3-}$ species on Co-Pi@CoP/NF and MnO$_2$@Co-Pi@CoP/NF are able to repel Cl$^-$ selectively and interfere less with the OH$^-$ adsorption; (3) simple immersion in KMnO$_4$ solution even yields a comparable amount of PO$_4^{3-}$ as the direct introduction of PO$_4^{3-}$ via the anodic treatment of CoP.

In situ Raman spectra of three electrodes were recorded at the potential window of 1.2 to 1.8 V vs. RHE. As shown in Fig. 4b, CoP/NF, Co-Pi@CoP/NF, and MnO$_2$@Co-Pi@CoP/NF all show two spectral bands corresponding to phosphorus-related species[32,33]. For the CoP/NF, a transformation from CoP to protonated PO$_4^{3-}$-related species due to the oxidation of P species occurs at a rather positive potential, with spectral bands centered at around 977.5 (symmetric stretching vibration, $\nu_{sym, P-O}$) and 1059.6 cm$^{-1}$ (asymmetric stretching vibration, $\nu_{asym, P-O}$)[33]. In contrast, both Co-Pi@CoP/NF and MnO$_2$@Co-Pi@CoP/ NF show such peaks in a wide potential range from 1.2 to 1.8 V vs. RHE due to the pre-existing Co-Pi layer. Also, such PO$_4^{3-}$-related Raman peaks of the latter two are stronger than those of the CoP/NF. Raman data thus imply that electrodes with more surface phosphates can achieve the PO$_4^{3-}$-based protection much earlier and over a wider range of applied potentials. A quantitative experiment was further performed to illustrate how the phosphate can mitigate the surface adsorption of Cl$^-$. The Co−Cl peak intensity was recorded in alkaline seawater with a gradually rising concentration of sodium phosphate using CoP/NF as an anode[55–57], and a concentration-intensity relationship can reflect the repelling Coulombic forces of surface PO$_4^{3-}$-related

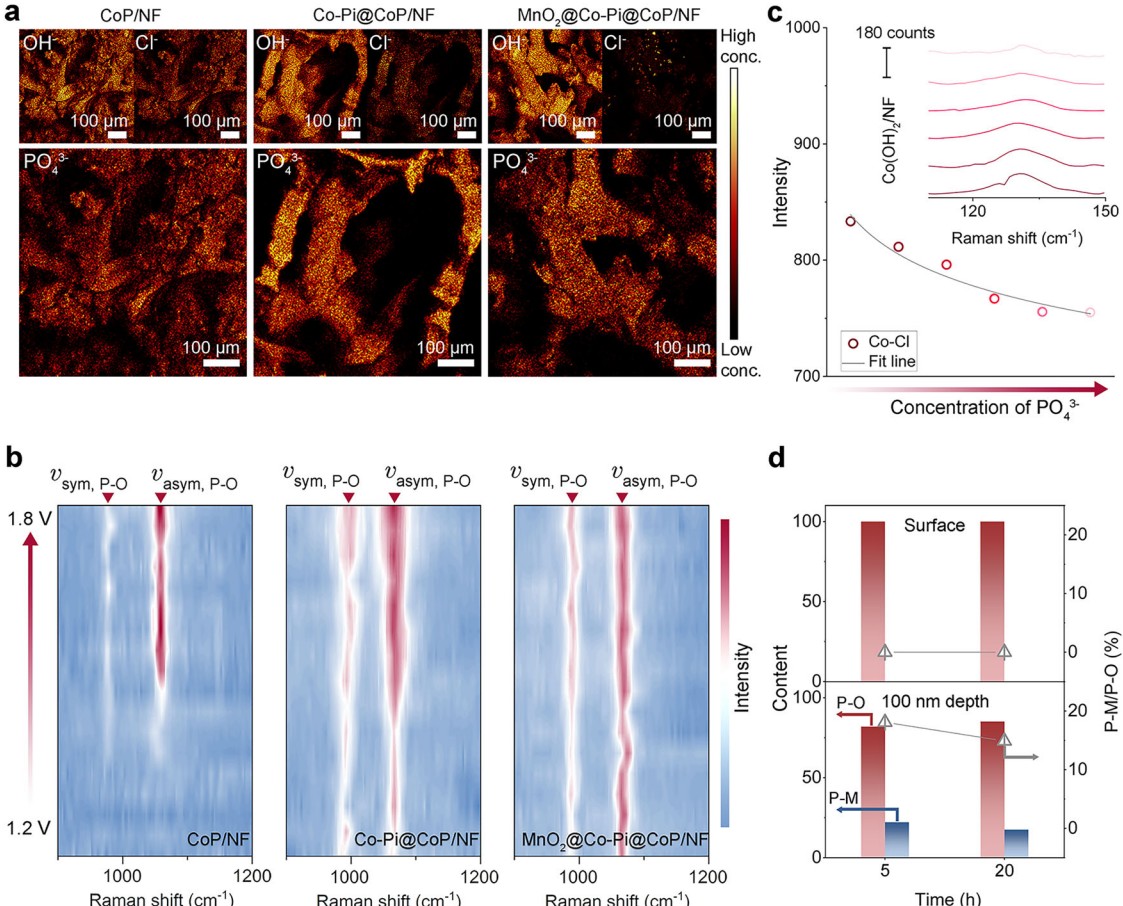

**Fig. 4 | 1st defense: $PO_4^{3-}$-enriched barriers. a** TOF-SIMS mapping of $OH^-$, $Cl^-$, and $PO_4^{3-}$ fragments from electrodes after 10-h eASO (conc. represents concentration). **b** Potential-dependent Raman spectra of the three electrodes recorded in alkaline seawater. The three graphs from left to right correspond to Raman spectra of CoP/NF, Co-Pi@CoP/NF, and MnO$_2$@Co-Pi@CoP/NF (sym and asym represent symmetric and asymmetric, respectively). **c** Co−Cl peak intensities as the function of the $PO_4^{3-}$ concentration. Inset image shows the $PO_4^{3-}$ concentration-dependent changes in the Co−Cl Raman peaks. **d** Specific changes of surface and internal phosphorus-related species after different electrolysis times. Source data are provided as a Source Data file.

species against $Cl^-$. A negative correlation is established between the Co−Cl peak intensity and the $PO_4^{3-}$ concentration in the electrolytes (Fig. 4c), indicating that the amount of chemically adsorbed chlorine atoms can decrease with the addition of $PO_4^{3-}$. Since MnO$_2$@Co-Pi@CoP/NF has a higher relative abundance of $PO_4^{3-}$-related anions, the first line of cation-selective defense can be established well. With regard to the depth and time spent in eASO electrolysis, there is a slight variation in the chemical state of the phosphorus species of the samples (Fig. 4d). An extended electrolysis duration from 5 h to 20 h only slightly lowers the relative proportion of P−M bonds in the interior (100 nm depth). Therefore, phosphorus species on the surface of MnO$_2$@Co-Pi@CoP constantly and primarily exist in the phosphate form, assisting the catalyst to filter out and reject the $Cl^-$. Meanwhile, XPS analysis of the Co 2$p$ region reveals a gradual decrease in the intensity of Co$^{δ+}$ related peak with increasing reaction time (Supplementary Fig. 27), indicating the progressive oxidation of CoP into hydroxide/oxide species. Despite this transformation, electrochemical impedance spectroscopy measurements show that the charge transfer resistance remains nearly unchanged before and after eASO with different reaction times (Supplementary Fig. 28), suggesting that the catalyst maintains its high electrical conductivity. This can be ascribed to the slow inward conversion of the CoP core, which continues to act as an efficient electron-conducting pathway, as well as the preserved one-dimensional nanowire structure that facilitates rapid electron transport.

## 2nd defense: well-dispersed nanosized γ-MnO$_2$

According to previous research findings from different groups, the electrodeposition of Mn-based oxides on IrO$_2$ improved the OER selectivity in seawater/saltwater[58–62]. In a recent work, γ- and δ-MnO$_2$ were verified to possess more CER active sites than those of α- and β-MnO$_2$, but γ-MnO$_2$ modified IrO$_2$ exhibited better ability to hinder CER, compared with α-, β-, and δ-MnO$_2$ counterparts[63]. Such results indicate that the design of eASO catalysts with fewer CER-active sites may not be the most suitable choice (note that CER itself is easy to occur). If the catalyst cannot activate even $Cl^-$, then the four-electron OER will be more difficult to occur. MnO$_2$@Co-Pi@CoP/NF exhibits enhanced OER activity, and the AIMD simulations confirm that the highest coverage of oxygen intermediates on the γ-MnO$_2$ is a key factor in achieving highly selective rejection of $Cl^-$, which is also consistent with the high OER-selectivity observed in the experiments. In addition, the open tunnel dimensions of γ-MnO$_2$ (e.g., 2.3 Å × 2.3 Å and 2.3 Å × 4.6 Å) are smaller than that of $Cl^-$ (3.7 Å), yet about the comparable diameters as water molecule (2.8 Å) and O$_2$ (2.92 Å)[64–66]. Therefore, this may also be one of the reasons why MnO$_2$ was employed as a blocking material with a semipermeable nature in the electrolytes with ample $Cl^-$.

Even with such understandings of the 2$^{nd}$ defense (i.e., MnO$_2$ in MnO$_2$@Co-Pi@CoP), relatively few research (1) modified MnO$_2$ on/within noble-metal free catalysts, (2) prepared MnO$_2$ without an electrodeposition step, and (3) were conducted in alkaline seawater

towards investigating how do Mn oxides contribute to more selective OER. Importantly, the Cl$^-$-blocking nature exhibited by MnO$_2$ is also not fully understood. Therefore, more characterizations with a specific goal to make the insights into the 2$^{nd}$ defense comprehensive and reliable were carried out. XPS data for post-eASO MnO$_2$@Co-Pi@CoP samples with and without Ar$^+$ etching preliminarily define the chemical states. Before eASO, the binding energy separation of the couple peaks ($\Delta E$) in Mn 3$s$ spectra (Fig. 5a) is 4.8 eV for the surface and 4.9 eV for the 100-nm depth. A higher $\Delta E$ typically represents a lower average oxidation state (AOS), and previously reported or standard MnO$_2$ typically possesses $\Delta E$ of 4.4–4.7 eV[37,38]. Thus, the AOSs for the interior and surface Mn species before eASO differ a little but both are below 4 (~3.5). The $\Delta E$ values of the catalyst surface after eASO are slightly lower than those of the interior, reflecting a higher surface oxidation state (~3.8) and a lower valence state of the interior (about 3.4 to 3.5). In comparison to the interior, electrolysis increases the AOSs of Mn species on the surface. Nevertheless, the rise of oxidation state shows an upper limit. AOSs after 20-h eASO stop increasing, and samples after 5 h and 100 h of testing almost have identical valence for both the surface and the 100 nm interior, which indicates that the oxidation state of the Mn oxides can gradually and self-adaptively stabilize during the reaction. The surface's Mn 2$p_{3/2}$ peak at 642.40 eV before eASO shifts to 642.60, 642.80, and 642.85 eV after 5-h, 20-h, and 100-h eASO (i.e., increased surface oxidation states), respectively, and the 100 nm depth's Mn 2$p_{3/2}$ peak at 640.85 eV before eASO shifts to 641.45 eV after 100 h eASO. This positive shift, both for surface and interior Mn species, toward higher binding energy is consistent with the formation of high-valence Mn species (e.g., Mn$^{4+}$). As shown in Fig. 5b, the Mn valence state fluctuations of the surface are greater relative to the fluctuations of the interior, suggesting that the reconstruction occurring at the surface is more intense. In addition, the spin energy gaps between Mn 2$p_{3/2}$ and Mn 2$p_{1/2}$ peaks show gap values of 11.8 eV (surface) and 11.9 eV (100 nm) before eASO. Electrolysis processes lead to decreased and almost equal gap values of 11.7 eV (surface) and 11.6 eV (100 nm) after 5 h eASO, 11.6 eV (surface) and 11.8 eV (100 nm) after 20 h eASO, and 11.6 eV (surface) and 11.8 eV (100 nm) after 100 h eASO. These values are reasonably close to those of previously reported MnO$_2$ (11.6–11.8 eV), implying almost dominant (near) surface Mn (IV) species after eASO[37–39]. It is important to note that the conclusion regarding the oxidation states derived from XPS data in the Mn 2$p$ and Mn 3$s$ XPS regions is not contradictory. Considering the sufficiency of XPS data, the oxidation state fluctuations of the samples are indeed very low, and the MnO$_2$ can be regarded as stable during the electrolysis. Despite the wide variety of sources, the corresponding signals in the O 1$s$ are of good value in analyzing changes in oxygen species for MnO$_2$@Co-Pi@CoP. The metal-oxygen bonds and absorbed oxygen species should be the dominant species, and binding energy shifts after eASO are reasonable (Supplementary Fig. 29) due to the complexity of amorphous phosphates and the randomness of MnO$_2$ distribution in them.

Before eASO, Raman spectrum of MnO$_2$@Co-Pi@CoP/NF only shows one peak at 651.5 cm$^{-1}$ that can be assigned to MnO$_2$ (Supplementary Fig. 2). Note that the MnO$_6$ octahedra determines the related Raman bands of MnO$_2$, and the band centered in the range of 640–680 cm$^{-1}$ corresponds to the symmetric stretching mode of the Mn−O bond (e.g., A$_{1g}$ mode of Mn−O vibrations) in MnO$_6$ octahedra and associated with the Jahn Teller distortion ($v_1$)[35,67–69]. As shown in Fig. 5c, the intensity of the $v_1$ peak after 5-h eASO is lower than that of the $v_1$ peak prior to the electrolysis, indicating a diminishment in the vibration mode and a possible Mn−O bond length decrease[67]. Moreover, the decrease in the distortion (e.g., Mn$^{3+}$) generally implies a shorter Mn−O bond length. Thus, the result is consistent with the elevated AOS (i.e., fewer Mn$^{3+}$) after 5-h eASO inferred from the XPS data. It is uncertain to know the degree of distortion based on the change in the oxidation state alone. This is because there are many

other possible factors, such as defects (e.g., oxygen vacancies), reconstruction, and adsorbed intermediates during the OER that may have an effect on the crystal structure and the degree of octahedral distortion. The impacts of these factors may be intertwined or counteracted by the impacts of valence changes. The $\Delta E$ data of the sample after 5 h eASO and the $v_1$ peak intensity change after 5 h eASO can cross-check the conclusion of a lower degree of distortion. Note that Mn oxidation states change only slightly after 20-h and 100 h electrolysis, which may not be enough to change the degree of octahedral distortion notably. Therefore, even if the intensity of the $v_1$ peak corresponding to 20 h testing is observed to be elevated compared to the intensity of the $v_1$ peak relating to 5 h of testing (Fig. 5c), it does not conflict with the slightly raised AOS. The $v_1$ peaks measured at different areas of electrodes after 100 h eASO are not much different from those of electrodes after 20 h eASO (Fig. 5c), which indicates that the structural changes of Mn species in MnO$_2$@Co-Pi@CoP have a limit and should be transitioned to a stable state during electrolysis. Note that the Raman spectral band region for Co species (300–640 cm$^{-1}$) partly overlaps with that for MnO$_2$. Thus, powder γ-MnO$_2$ catalysts were prepared for the Raman spectroscopy measurements in electrolytes containing different anions (Fig. 5 d), and Raman data that correspond to γ-MnO$_2$ were analyzed to reveal the interaction between ions and this γ-MnO$_2$ under anodic potentials. The electrolytes are 1 M KOH solution, 1 M KOH solution with extra PO$_4^{3-}$ (0.25 M and 0.5 M), KOH solution with extra Cl$^-$ (0.25 M and 0.5 M), and KOH solution with both PO$_4^{3-}$ (0.25 M and 0.5 M) and Cl$^-$ (0.5 M), representing four different situations. Except for $v_1$ peaks, the other two major peaks of MnO$_2$ are observed at 575.8 cm$^{-1}$ ($v_2$) and 523.2 cm$^{-1}$ ($v_3$) under each condition[69]. As shown in Fig. 5d, applying an anodic potential of 1.5 V to MnO$_2$@Co-Pi@CoP in an aqueous KOH solution without PO$_4^{3-}$ or Cl$^-$ does not cause a noticeable change in the ratio of the $v_2$ bond intensity/$v_1$ bond intensity ($I_{v2}/I_{v1}$), which reflects the high ordering of MnO$_6$ octahedra[69]. Noticeably, the addition of Cl$^-$ alone or PO$_4^{3-}$ alone to KOH solution can cause diametrically opposed structural changes in the MnO$_6$ octahedra. As the concentration ratio of Cl$^-$/OH$^-$ rises from 0 to 0.25 and then to 0.5, the $I_{v2}/I_{v1}$ increases from 1.11 to 1.14 and then to 1.16. In contrast, by increasing the PO$_4^{3-}$/OH$^-$ concentration ratio from 0 to 0.25 and subsequently to 0.5, the $I_{v2}/I_{v1}$ falls from 1.11 to 1.06 and finally to 1.02 (Fig. 5e). Therefore, the observed increase and decrease trends of the $I_{v2}/I_{v1}$ values may be suggestive of opposite structural changes due to PO$_4^{3-}$ and Cl$^-$. In addition, the presence of Cl$^-$ will reduce the interaction of PO$_4^{3-}$ with the catalyst, further indicating that the two ions may have opposite effects on the structural changes of MnO$_2$.

TEM image of MnO$_2$@Co-Pi@CoP after eASO shows a well-maintained 1D structure, and well-defined crystal with an interlayer spacing of 0.233 nm is observed from the HRTEM image, corresponding to the (230) plane of γ-MnO$_2$ (Fig. 5f). Even though the operation at high anodic $j$ can produce a multitude of amorphous oxyhydroxides, and amorphous Co species (e.g. CoOOH) were derived from the surface of the post-electrolysis material, XRD patterns of MnO$_2$@Co-Pi@CoP/NF after 5 h eASO, 20-h eASO, and 100-h eASO (Supplementary Fig. 30) still show diffraction peaks of γ-MnO$_2$. This reflects the high stability of the catalyst. Due to the interference of Ni peaks, powder samples were used for in situ XRD characterization. Electrolysis at varying applied potentials in alkaline seawater does not produce any observable or significant change in the intensity as well as the position of diffraction peaks of γ-MnO$_2$ (Fig. 5f). This again indicates a high level of stability of γ-MnO$_2$ during electrolysis even under high $j$. The protective effects of the amorphous Co-Pi or amorphous CoOOH surrounding the γ-MnO$_2$ nanoparticles may be one of the causes of the high degree of stability. The long-range disordered structure should change the speed and path of the reactants, like ions and molecules, to reach the γ-MnO$_2$ and prevent exterior substances from making direct contact with the internal crystalline γ-MnO$_2$. The

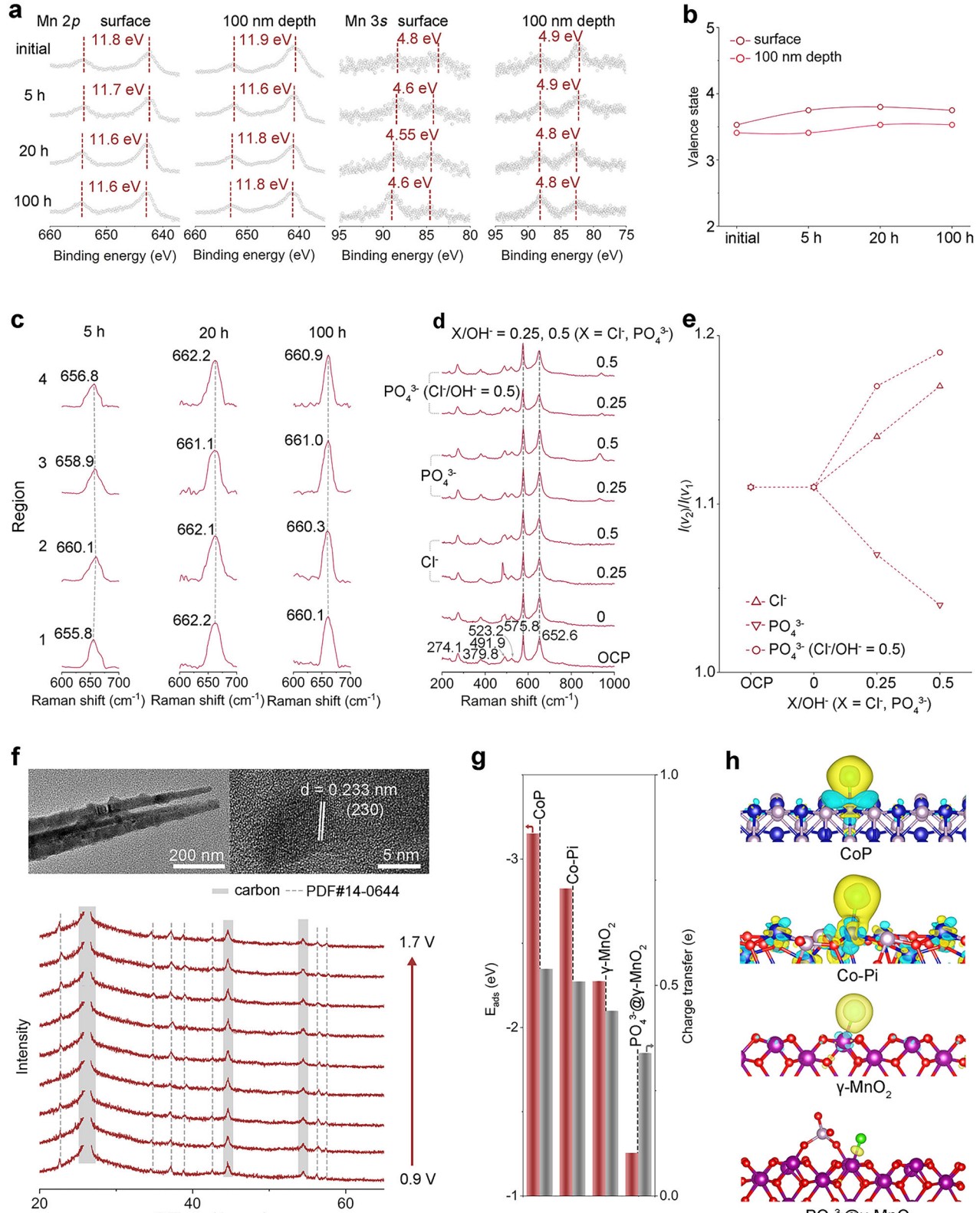

**Fig. 5 | 2nd defense: well-dispersed nanosized γ-MnO₂. a** XPS depth profiles of MnO₂@Co-Pi@CoP before and after eASO. **b** The average valence state changes of Mn calculated from data in 3s spectra. **c** Raman spectra showing changes in the $v_1$ band for MnO₂@Co-Pi@CoP/NF electrodes after 5 h, 20 h, and 100 h eASO. **d** Raman spectra recorded in a series of alkaline electrolytes with different anions

(OCP: KOH solution) for powder γ-MnO₂ loaded on carbon paper. **e** $I_{v2}/I_{v1}$ values varying with anion content and species. **f** TEM and HRTEM images of MnO₂@Co-Pi@CoP after electrolysis and in situ XRD patterns for powder γ-MnO₂. **g** $E_{ads}$ and charge transfer between Cl⁻ and different materials. **h** Differential charge density diagrams of four catalyst models. Source data are provided as a Source Data file.

diffusion process in the amorphous structures changes the original energy and the pathway, which allows the molecules or ions to contact the γ-MnO$_2$ in a milder way, lowering the risk of structural damage. Moreover, the amorphous structures may block harmful impurities in seawater, avoiding some undesirable side reactions.

Based on the analyses of the experiments and AIMD simulations, density functional theory (DFT) calculations were performed to acquire a comprehensive understanding of the role that the PO$_4^{3-}$@γ-MnO$_2$ plays. Adsorption energy (E$_{ad}$) of Cl$^-$ was evaluated to reveal the interaction between Cl$^-$ and different catalytic materials (Fig. 5g). PO$_4^{3-}$@γ-MnO$_2$ (100) exhibits the weakest interaction with Cl$^-$ with an E$_{ad}$ of −1.26 eV, whereas γ-MnO$_2$ (100) has an E$_{ad}$ of −2.28 eV, CoP (011) has an E$_{ad}$ of −3.15 eV, and Co-Pi (001) has an E$_{ad}$ of −2.83 eV. The differential charge density diagrams (Fig. 5h) reveal that charge transfers of 0.44 e, 0.54 e, and 0.51 e occur from γ-MnO$_2$ (100), CoP (011), and Co-Pi (001) to Cl$^-$, respectively, whereas this value is only 0.34 e between PO$_4^{3-}$@MnO$_2$ and Cl$^-$. Notably, *Cl species cannot be generated on the pristine CoOOH (001) surface (Supplementary Fig. 31) during the structure relaxation, so the decrease in catalytic performance of CoOOH should be attributed to the poisoning of reconstructed Co sites by Cl$^-$ (Supplementary Fig. 26d).

### 3rd defense: cage-like nanoarchitecture

While advanced electrodes, like self-supported ones, have been widely developed for seawater electrolysis applications, far fewer work-optimized nanostructures in terms of their microscale arrangement[70]. Differences in the interfacial mass transfer-induced forces, the bubble distribution, and forces of bubbles on structures due to nanowire arrangement differences are thus compared. Two models were built by summarizing the structural features from CoP and MnO$_2$@Co-Pi@CoP (Fig. 6a and Supplementary Fig. 32), and the densities, Young's modulus, and Poisson's ratio values of powder samples (CoP and MnO$_2$@Co-Pi@CoP) are provided in Supplementary Table 4. The contour maps of the velocity field with cross-section views illustrate that disorderly arranged nanowires (i.e., the contrast sample) lead to higher velocity gradients at the bottom and middle regions of the nanowire with a maximum velocity of about 0.1336 m s$^{-1}$ (Fig. 6b), while the optimal sample has a more uniform overall velocity distribution with a maximum velocity of about 0.1373 m s$^{-1}$ (Fig. 6c), which is higher than that of the contrast sample. Such a higher uniformity of velocity results from a much more regular arrangement of tip-connected nanowires. In contrast, the contrast sample exhibits a small velocity gradient in the overlap region due to the disordered nanowire arrangement. Also, a region of lower velocity is observed at the overlap areas, which is not favorable for the bubble detachment.

Gas bubble distribution maps demonstrate that bubbles on the disorderly arranged nanowires are mainly generated and detached at the bottom and side wall surfaces (Fig. 6d), while the bubbles of the optimal sample are generated and detached at the bottom, the surface bumps, and the tip/head part (Fig. 6e). Also, the bubbles distribute evenly on the optimal sample as well, with more notable generation and detachment processes. We point out that the rougher surfaces of MnO$_2$@Co-Pi@CoP, represented by surface bumps in the corresponding model, reduce the adhesion of bubbles, and the curved tip-connected structure can allow the bubbles to detach from the nanowire more smoothly. The micro-arrangement of the disordered nanowires is comparatively messy, and the resulting efficiency of bubble generation and detachment is relatively slower. Thus, the contrast sample suffers from uneven distributions of the velocity field and gas bubbles, causing lower efficiency of bubble release. The tip-connected cage-like nanoarray sample should be more advantageous in facilitating the bubble transport and detachment, which is verified further through experiments. First of all, the trend of gas bubble contact angles (Supplementary Fig. 33) is as follows: CoP/NF (38.3°) > Co-Pi@CoP/NF (28.1°) > MnO$_2$@Co-Pi@CoP/NF (10.4°). Thus, the

improved aerophobic and hydrophilic surface properties of MnO$_2$@Co-Pi@CoP/NF can facilitate O$_2$ bubble release during the eASO process, especially at high $j$. In addition, operando single high-frequency was applied to three electrodes to evaluate their bubble release behaviors during the eASO. As shown in Supplementary Fig. 34a–c, the minimal changes in the resistance (Z$_s$) over time of MnO$_2$@Co-Pi@CoP/NF at a $j$ of 100 mA cm$^{-2}$ indicate bubbles generated at such $j$ can be released promptly, whereas CoP/NF and Co-Pi@CoP/NF both show large fluctuations in Z$_s$ under identical testing conditions. At a higher $j$ of 500 mA cm$^{-2}$, sharp fluctuations in Z$_s$ occur on both CoP/NF and Co-Pi@CoP/NF, but no significant changes of Z$_s$ are achieved by MnO$_2$@Co-Pi@CoP/NF. As $j$ continues to increase to 1000 mA cm$^{-2}$, the advantage of the cage-like nanoarray of MnO$_2$@Co-Pi@CoP on NF still allows a considerably more uniform bubble release process, as evidenced by the consistent and smaller variations in Z$_s$ curves. The related fast Fourier transform curves at 1000 mA cm$^{-2}$ further reveal the smallest amplitudes for MnO$_2$@Co-Pi@CoP/NF, indicating much easier and faster detachment of surface-bound gas bubbles (Supplementary Fig. 34d). Furthermore, bubble adhesion force measurements on different samples were conducted, as shown in Supplementary Fig. 35. MnO$_2$@Co-Pi@CoP/NF exhibits the lowest bubble adhesion force (−6.2 µN) due to its distinctive cage-like nanoarray structure, compared to CoP/NF (~10.6 µN) and Co-Pi@CoP/NF (~10.4 µN). This low adhesion force promotes more efficient bubble detachment, where timely bubble release helps maintain the exposure of active sites to the electrolyte and reduce stress, thereby minimizing activity degradation during long-term eASO.

The force distribution analysis of bubbles on the nanowires yields key information. First of all, the pressure of the disorderly arranged nanowires is concentrated at the bottom (the red region in Fig. 6f). The local zoomed-in image clearly demonstrates the centralized stress in the localized region at the bottom, with a pressure maximum of about 8.223 Pa. Note that the tip-connected sample has two main stress concentration regions (zoomed-in images in Fig. 6g): the tip and the bottom parts with a maximum localized stress of about 5.726 Pa (only 69.6% of the maximum localized stress for the contrast sample). The decreased localized stress for the tip-connected model indicates improved mechanical stability in response to the bubble attack during electrolysis, which is a significant advantage.

In addition, another model was constructed to vary the number of nanowires within the cage-like structure (Supplementary Fig. 36a). This model still exhibits an improved velocity field homogeneity (Supplementary Fig. 36b), an even distribution of bubbles (Supplementary Fig. 36c) and better mechanical performances (Supplementary Fig. 36d). For instance, the maximum localized stress for this model is about 6.137 Pa, which is still lower than 8.223 Pa for the model with a disordered catalyst arrangement. All simulation results confirm that the enhanced mechanical properties for MnO$_2$@Co-Pi@CoP/NF are mostly attributed to the nanocage-like architecture of tip-connected nanowires.

## Discussion

We demonstrate a triple-protected monolithic catalyst that realizes little corrosion during 3000 h of alkaline seawater electrolysis under ampere-level $j$. Three key defense designs include Co-Pi, MnO$_2$, and the cage-like nanoarray. PO$_4^{3-}$-enriched barriers and well-dispersed nano-sized MnO$_2$ together alleviate low-selectivity-induced corrosion by preferentially filtering out Cl$^-$ as two semipermeable blocking materials. Cage-like architectures composed of tip-connected nanowires with rough surfaces effectively minimize physical damages (i.e., tensile and vibrational forces) from bubble escaping/collapsing, thus achieving substantially improved bubble release behaviors and much lower localized stresses. This work not only provides a top-level noble metal-free electrocatalyst for seawater oxidation processes but also presents guiding anode design strategies with the triple defense concept towards simultaneously solving two long-standing issues that jointly cause poor

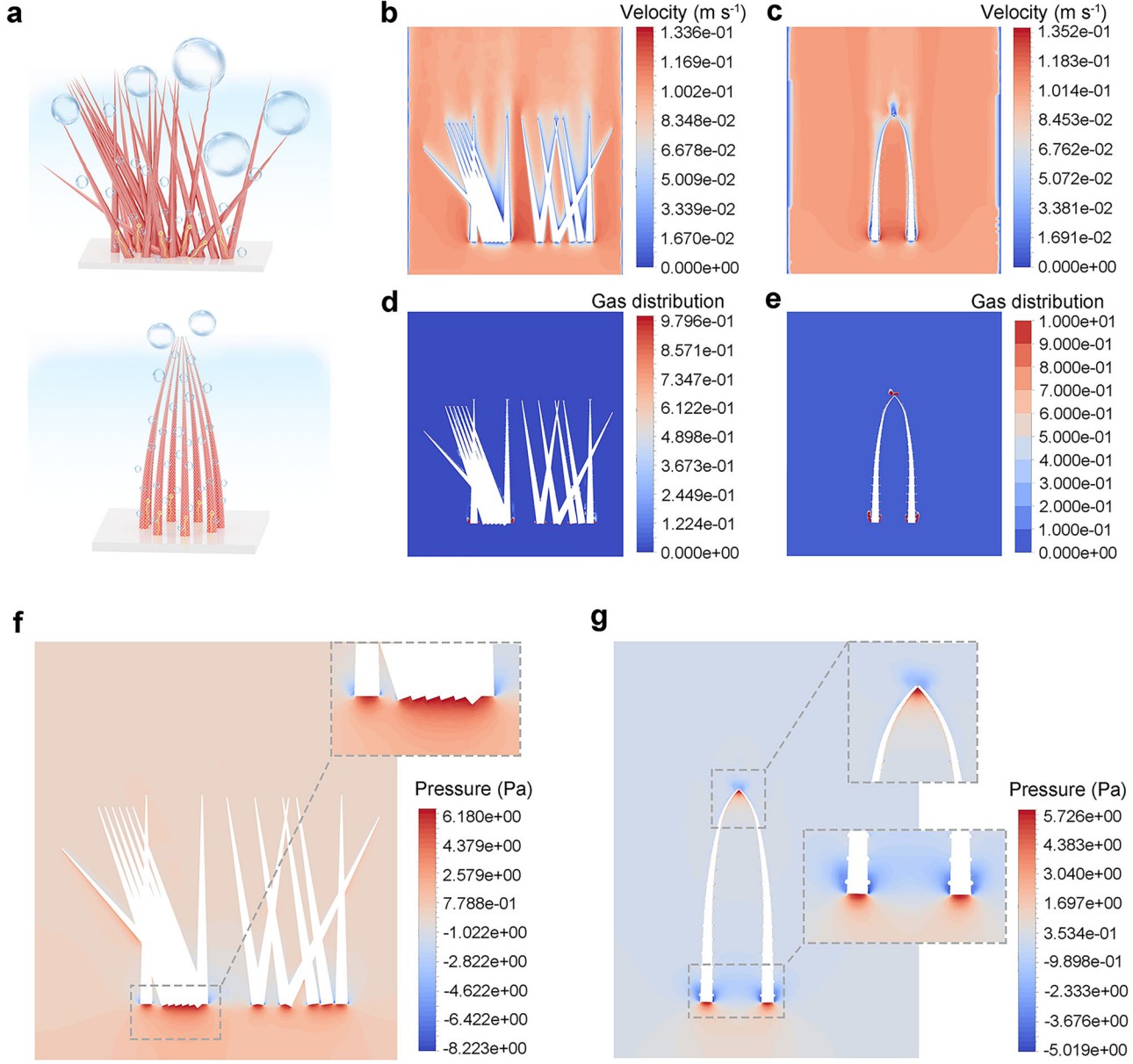

**Fig. 6 | 3rd defense: cage-like nanoarchitecture. a** Schematic illustration of the typical arrangements of nanowires and the special tip-connected arrangement of nanowires. **b** Velocity field for the contrast structure. **c** Velocity field for the optimal structure. **d** Gas bubble distribution on the contrast structure. **e** Gas bubble distribution on the optimal structure. **f** Distribution of forces on the contrast structure from the bubble. **g** Distribution of forces on the optimal structure. Source data are provided as a Source Data file.

anode stability in seawater electrolysis: chlorine species-induced chemical corrosion and physical damages from external forces.

## Methods
### Materials
Cobaltous nitrate hexahydrate ($Co(NO_3)_2 \cdot 6H_2O$, AR 99%), ammonium fluoride ($NH_4F$, AR 98%), urea ($CO(NH_2)_2$, AR 99%), sodium hypophosphite ($NaH_2PO_2$, AR 98%) potassium hydroxide (KOH, 95%), hydrochloric acid (HCl), ruthenium oxide ($RuO_2$), Pt/C (20 wt.%), Nafion (5 wt.%), and tetramethylammonium hydroxide (TMAOH, 97%), all of which were obtained from Aladdin Ltd. in Shanghai. Sodium hypochlorite (NaClO) was obtained from Beijing Chemical Corp. Nickel foam (0.2 mm thickness) was obtained from Qingyuan Metal Materials Co., Ltd. Natural seawater was sourced from Huangdao District, Qingdao, and ultrapure water (18.3 MΩ·cm) was used in all experiments. All chemicals were used without further purification.

### Synthesis of CoP/NF
A piece of cut and shaped NF (2*3 cm²) was immersed in acidic solution (3 M HCl) for a high-intensity ultrasound treatment for at least 15 min. Subsequently, the NF with slightly etched surfaces was ultrasonically cleaned repeatedly by alternately using deionized water and anhydrous ethanol. Co(OH)F/NF was prepared hydrothermally, and 0.485 g $Co(NO_3)_2 \cdot 6H_2O$, 0.15 g $NH_4F$, and 0.5 g $CO(NH_2)_2$ were dissolved in water (33 mL) under vigorous stirring (>20 min stirring and 15 min ultrasound treatment) to prepare the reaction medium. Then, the medium together with the slightly etched NF was transferred into an autoclave to grow Co(OH)F during a 6-h-long reaction, and the temperature required for this process was 120 °C. After naturally cooling down, the precursor was taken out and carefully rinsed with water and ethanol. Then, this precursor was dried in a specific temperature environment (around 70 °C). CoP/NF was obtained by placing 1 g of $NaH_2PO_2$ and the precursor in two porcelain boats at the upper and

lower ends of a tube furnace and holding the temperature at 300 °C for 2 h. CoP/NF shows a loading of ~2.1 mg cm$^{-2}$.

### Synthesis of Co-Pi@CoP/NF

A three-electrode system (i.e., a polished graphite rod, an Ag/AgCl electrode, and a piece of CoP/NF) was adopted for electrode synthesis. A 40 mL solution of 0.5 M Na$_3$PO$_4$ was prepared for introducing surface phosphates. The CoP/NF electrode was polarized at 1.1 V vs. Ag/AgCl for 7 min, and the obtained electrode was rinsed with water and ethanol, and further dried at around 70 °C for 12 h. Co-Pi@CoP/NF shows a loading of ~2.3 mg cm$^{-2}$.

### Synthesis of MnO$_2$@Co-Pi@CoP/NF

MnO$_2$@Co-Pi@CoP/NF was obtained by soaking CoP/NF (size: 1*2 cm$^2$) in 0.5 M KMnO$_4$ solution for 45 min. It shows a loading of ~2.7 mg cm$^{-2}$.

### Synthesis of commercial counterparts

The formula of the catalyst ink includes 20 mg of RuO$_2$ (or Pt/C), 490 μL of water, 490 μL of ethanol, and 20 μL Nafion binder. A uniform ink was obtained after a long period of ultrasonic treatment (over half an hour). 125 μL of ink was then slowly dripped onto a slightly etched NF (1*1 cm$^2$) with a micropipette and simultaneously dried. The mass loading is about 2.5 mg cm$^{-2}$.

### Characterizations

The morphology and micro-structure information of samples were revealed via SEM (ZISS 300), TEM (JEM-F200, JEOL Ltd.), and HRTEM (FEI Tecnai G2 F20). XRD (Shimadzu XRD-7000 diffractometer with Cu Kα radiation (40 kV, 30 mA) and XPS (ESCALABMK II X-ray photoelectron spectrometer) measurements were used for chemical structure and phase components characterization. The XRD measurement was performed with self-supported electrodes with a size of around 1*1 cm$^2$. The XPS measurement used samples with a size of around 0.25*0.25 cm$^2$, and all the samples needed to be pressed. The depth XPS measurement used high-purity Ar gas and an etching time of 37 s. The sample size for Raman spectroscopy measurement (with 532-nm laser and an attenuation degree of 10%) was 0.5*0.25 cm$^2$, the attenuation degree was 10% and the acquisition time was 20 s.

### Electrochemical measurements

Most of the electrochemical assessments were conducted on a CHI660b workstation. Seawater will be treated with sodium carbonate and KOH to obtain alkaline seawater (1 M KOH + seawater, pH = 13.95 ± 0.02) before the test. The stability test was evaluated utilizing a LAND CT2001A workstation. The electrodes' performance was scrutinized in alkaline seawater with a conventional three-electrode configuration, utilizing a cylindrical graphite rod electrode, a Hg/HgO electrode, and a working electrode (e.g., MnO$_2$@Co-Pi@CoP/NF). Except for methods, all potentials reported in our work were converted to RHE scale based on the equation:

$$E_{RHE} = E_{Hg/HgO} + 0.059 \times pH + 0.098V \qquad (1)$$

The iR-compensated potential was obtained after correcting the solution resistance measured according to the equation:

$$E_{corr} = E - iR \qquad (2)$$

where E is the original potential, R is the solution resistance, i is the corresponding current, and $E_{corr}$ is the iR-compensated potential.

### Catalyst activation and polarization data

Catalysts were activated with a CV technology in alkaline seawater (1 M KOH + seawater) in the potential range of 0–1 V vs. Hg/HgO (20 cycles,

scan rate: 20 mV s$^{-1}$). The activated sample (e.g., MnO$_2$@Co-Pi@CoP/NF) was evaluated by using the LSV technique at a scan rate of 5 mV s$^{-1}$.

### FTacV characterization

The applied potential intervals were initially set from 0 to 0.8 V vs. Hg/HgO, which were expanded by 0.1 V at each time to collect the data. The sixth harmonic data were processed and analyzed.

### Impedance spectroscopy characterization

Potential-dependent impedance spectroscopy experiments were performed in the potential range of 100000–0.01 Hz, and the data were recorded every 40 mV increase from open circuit potential (OCP).

### TOF calculation

The potential range for CV was 0–1.2 V vs. Hg/HgO. According to the voltammetry data, the linear regression equation of oxidation peak currents and scan rates can be obtained to further calculate the slope in the equation:

$$slope = n^2F^2m/4RT \qquad (3)$$

where $n$ stands for electron transfer number, and R and T are the ideal gas constant and absolute temperature. Then, m as the number of active sites can be calculated.

According to the equation:

$$TOF = jA/4Fm \qquad (4)$$

where $A$ is the geometric area size and F is Faraday constant. TOF values of different electrode materials can be finally obtained. We performed the TOF test three times and used the median value for subsequent calculations.

### Activation energy calculation

LSV scans were performed at 30, 40, and 50 °C with a scan rate of 5 mV s$^{-1}$ for various electrode materials. $j$ and the corresponding temperatures under a certain potential were plotted to get the slope, and then the activation energy was obtained based on Arrhenius formula.

### Methanol oxidation experiment

LSV tests were carried out in alkaline seawater and alkaline seawater containing 0.5 M methanol, respectively, with a scan rate of 5 mV s$^{-1}$.

### pH-changing reactions

Natural seawater treated with Na$_2$CO$_3$ was diluted to obtain alkaline seawater with pH values of 12.5, 13, 13.5, and 14. A three-electrode system was used to record the data under different pH.

### Tetramethylammonium cations (TMA$^+$)-based Raman test

Replacing K$^+$ with TMA$^+$ can study the effects of cations on reactions. The MnO$_2$@Co-Pi@CoP/NF electrode was tested by LSV scans in seawater solution with 1 M KOH or 1 M TMAOH at 1.0 V vs. Hg/HgO for 300 s, then washed with water and ethanol for Raman characterizations. The parameters were the same as for the ex situ Raman tests.

### In situ Raman characterization

Calibrations were carried out with silicon, based on a standard peak of 521.7 cm$^{-1}$. Then, the electrode samples were attached to copper tape and further assembled into a cell. Add about 16 mL of alkaline seawater electrolyte and keep it circulating using a pump. A Pt wire, an Hg/HgO electrode, and the self-supported electrode were employed to record the Raman data. Data at different potentials were collected several times during the reaction and focused with a 50x telephoto lens. The electrolysis time at each potential was 400 s. The power attenuation level was 20% and the testing time was 20 s.

## Electrochemical mass spectrometry characterization

The electrodes were cut to a length of 1 cm and a width of 1 cm and placed in the corresponding electrochemical in situ cell. Then, alkaline seawater solution containing $H_2^{18}O$ was slowly fed into the cell. After that, the oxygen exchange was carried out by CV technology at a potential range of 0–1 V vs. Hg/HgO (20 cycles) with a scan rate of 20 mV s$^{-1}$. Following the oxygen exchange process, the electrode was taken out and then subjected to a rinsing procedure. Mass spectrometry signals were collected in alkaline seawater.

## Anion exchange membrane-based electrolyzer

A piece of pretreated FAA-3-PK-75 was used to transport anions. The membrane was soaked in a saturated NaCl solution for 24 h, then rinsed with water. This membrane was stored in water for short periods when not in use. The electrolysis test was carried out with CHI 1140c workstation. The ink formulation for the catalyst layer of the cathode contained 20 mg of commercial Pt/C, 490 μL of water, 490 μL of anhydrous ethanol, and 20 μL of Nafion. The ink was evenly dispersed after a long time of ultrasound treatment. Then, 250 μL of ink was dropped on a piece of HCl-treated NF (1*1 cm$^2$) several times with a micropipette and dried under a heating lamp. The electrolyzer was assembled with the Pt/C/NF as the cathode and a piece of MnO$_2$@Co-Pi@CoP/NF. The stability data of the electrolyzer were recorded using a LAND CT2001A workstation.

## Dynamic single frequency impedance

Different samples were tested at the frequency corresponding to the phase angle of 1 and were tested at the potentials corresponding to the $j$ of 100, 500, and 1000 mA cm$^{-2}$. The test time was set at 1000 s.

## Calculation methods

DFT calculations and AIMD simulations both reveal the underlying mechanisms. The exchange-correlation interactions were described by the revised Perdew-Burke-Erzerhof (rPBE) functional with the projector-augmented wave (PAW) approximation. The view interaction was considered by using the DFT-D3 method. The plane wave cutoff energy was set to 400 and 300 eV for structural optimization and AIMD, respectively. Monkhorst-Pack k-point was used to sample the Brillouin zone. The energy convergence criterion was set to 1×10$^{-5}$ eV, while the force convergence criterion was −0.02 eV Å$^{-1}$. The adsorption energy was calculated with the following equation:

$$E_{ad} = E^*_a - E^* - Ea \qquad (5)$$

where $E_{ad}$ is the adsorption energy, $E^*_a$, $E^*$, and $E_a$ are energies of the surface of catalyst with adsorbate, the pristine surface of catalyst, and adsorbate, respectively.

For the AIMD simulations, the target temperature was set to 300 K by a canonical ensemble condition (NVT) with a Nose-Hoover thermostat, and the whole system contains 45 H$_2$O, one K$^+$, one OH$^-$, and one Cl$^-$ to simulate the alkaline environment. Each system was subjected to an equilibrium period of 3 ps to equilibrate the structure and construct the electrochemical interface close to reaction condition. If the OH$^-$ diffuses to the surface to form *OH species or the H of *H$_2$O species diffuses to the OH$^-$ to form H$_2$O and *OH species during equilibrium period, another OH$^-$ anion and H$_2$O molecule were added to the system. This process was repeated until no new *OH species formed on the surface and the OH$^-$ in solvent could exist stably. To facilitate structural visualization, the optimized electronic structure is provided in Supplementary Data 1.

## Simulation methodology details

The main purpose of the simulation study is to analyze the effect of external forces on the optimal sample representing MnO$_2$@Co-Pi@CoP on NF and two contrast samples. The corresponding models are included in Supplementary Data 1. The velocity field, the bubble distribution, and the force distribution of gas bubbles on the catalyst structures were studied by constructing the geometric models of the optimal sample and contrast samples, combined with computational fluid dynamic simulations. Polyhedral mesh and hexahedral mesh were used to divide the sample, and the results were compared and analyzed to evaluate the mechanical properties of the optimal sample under the shock of external forces from bubbles. The simulation was based on four prerequisites. Firstly, different nanowire structures were simplified to more regular geometric models to facilitate the simulation. Secondly, the fluid was an incompressible fluid, and the interaction between the bubble and the fluid was a quasi-stable process. Thirdly, the wall surface is smooth without slip, and the bubble distribution and movement follow the influence of a unidirectional flow field. Lastly, the temperature field does not notably affect the hydrodynamic behavior, so the coupling of thermal effects was ignored.

## Hydrodynamic equations

Reynolds mean Navier-Stokes (RANS) equation and shear-stress transport (SST) k−ω turbulence model were used in hydrodynamic simulation to describe the turbulence characteristics. Three main governing equations include the continuity equation and momentum equation, and these equations are given one by one as follows:

$$\frac{\partial \rho}{\partial t} + \nabla \cdot (\rho v) = 0 \qquad (6)$$

$$\rho \left( \frac{\partial v}{\partial t} \right) + v \cdot \nabla v = -\nabla p + \nabla \cdot \tau + f \qquad (7)$$

where $f$ represents external force. This equation describes the momentum change of a fluid, taking into account the action of inertial forces, pressure gradients, viscous forces, and external forces.

$$\rho c_p \left( \frac{\partial T}{\partial t} + v \cdot \nabla T \right) = k \nabla^2 T + \Phi \qquad (8)$$

where $c_p$ is specific heat at constant pressure, $T$ represents temperature, $k$ represents heat conductivity, and $\Phi$ represents a dissipative function.

## Bubble dynamics equations

The motion of bubbles in the flow field was described by the discrete phase model (DPM), which mainly includes the following equations:

$$m_p \frac{du_p}{dt} = F_d + F_g + F_b + F_l + F_t \qquad (9)$$

where $F_d$ is drag force (caused by the velocity difference between the fluid and the bubble), $F_g$ is gravity force (determined by the mass of the bubble and the acceleration of gravity), $F_b$ is buoyancy force (related to the volume of fluid displaced by bubbles), $F_l$ is lift force (produced by the effect of fluid rotation), and $F_t$ is additional mass force (the effect of fluid acceleration on the bubble was taken into account). These forces together determine the trajectory and acceleration of the bubble in the flow field.

## Turbulence model

The SST k−ω model, which combines the advantages of the standard k−ω model and the k−ε model, has high precision in the boundary layer region and is suitable for separation flow. The turbulent kinetic energy equation and turbulent vorticity equation used are shown below:

$$\frac{\partial(\rho k)}{\partial t} + \nabla \cdot (\rho k v) = P_k - \beta^* \rho k \omega + \nabla \cdot [(\mu + \sigma_k \mu_t) \nabla k] \qquad (10)$$

$$\frac{\partial(\rho\omega)}{\partial t} + \nabla \cdot (\rho\omega\mathbf{v}) = \frac{\gamma}{v_t}P_k - \beta\rho\omega^2 + \nabla \cdot \left[(\mu + \sigma_\omega\mu_t)\nabla\omega\right] \\ + 2(1 - F_1)\rho\sigma_\omega\frac{1}{\omega}\nabla k \cdot \nabla\omega \tag{11}$$

where $P_k$ is turbulence generation, and $F_1$ is a hybrid function for switching inside and outside the boundary layer, model coefficients are $\beta = 0.075$, $\beta^* = 0.09$, $\gamma = 0.52$, $\sigma_k = 0.5$, and $\sigma_\omega = 0.856$. The SST k-ω model is able to provide better prediction accuracy in the high Reynolds number region while accurately capturing turbulence characteristics in the near-wall region.

### Boundary conditions

In the geometric region division, the inlet boundary was set as a velocity inlet with an inlet flow rate of $0.1\,\text{m s}^{-1}$. The outlet boundary was set as a pressure outlet with a pressure of 0 Pa (relative pressure). A no-slip condition was used for the wall boundary (assuming no significant effect on the fluid motion). For the bubble parameter, the initial bubble velocity was consistent with the inlet flow rate.

### Grid setup and quality

Regarding the meshing methodology, the optimal sample utilized a polyhedral mesh, generating a total of 662,496 cells with a mesh quality ranging from 0.190 to 0.997 and an average mesh quality of 0.942. One of the contrast samples (optimal sample 2) uses a hexahedral grid with 2,989,056 cells and a grid mass ranging from 0.04 to 1 with an average mass of 0.706. Another contrast sample was generated using a polyhedral mesh with a total of 651,721 cells, with a mesh quality ranging from 0.180 to 0.998 and an average mesh quality of 0.953. For the grid independence validation, the tests were performed at different grid densities, and the validation results showed no significant difference in convergence with the number of grids.

## Data availability

The source data generated in this study are provided in the Source Data file. Source data are provided with this paper.

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

## Acknowledgements

This work was supported by the Free Exploration Project of Frontier Technology for Laoshan Laboratory (No. 16-02). F.G. would like to thank the financial support from Science and Technology Department of Jiangsu Province (SBZ2024080168 and SBT2024030057).

## Author contributions

X.S., Z.L., and J.L. designed the research and wrote the manuscript. J.L. conceptualized the paper. Z.L., M.Z., and S.S. synthesized catalysts and conducted experiments. Z.L. and J.L. analyzed the data. Z.L., Y.R., Z.C., C.Y., H.W., Y.L., and S.L. completed the schematic and data drawing. S.H. and F.G. conducted theoretical calculations. Y.Y., X.S., and B.T. guided the work. All authors discussed the experimental results.

## Competing interests

The authors declare no competing interests.
