## [Transparent Peer Review file · Nature Communications]

A triple-defense electrocatalyst for robust seawater oxidation

Corresponding Author: Professor Xuping Sun

Version 0:

Reviewer comments:

Reviewer #1

(Remarks to the Author)

The authors pursue integrating a triple defense at the nanoscale to enhance chemical and mechanical lifespans of anode catalysts, which is impressive. In fact, it is still rare to simultaneously optimize both the mechanical and chemical stability of an electrode in a highly corrosive environment such as seawater. This makes the findings of this work interesting and of great universal significance. In situ and ex situ data nicely visualize and reveal the triple-defense mechanisms, indeed ensuring the reliability. Therefore, this work represents a solid contribution to the field and should be accepted with minor revisions.

1. This paper points out that Co-Pi can promote the generation of active sites, which is an important point for understanding the improvement of performance. A better characterization of the Co-Pi coating depth on CoP is essential, necessitating transmission electron microscope images to reveal the structure.

2. From the results presented in Figure 3, it is evident that both manganese dioxide and phosphate ions contribute to the enhanced stability of cobalt phosphide under the current density of 2 A cm^{-2} . Specifically, what is the quantitative contribution of manganese dioxide to the stability improvement? Additionally, can manganese dioxide alone significantly enhance the long-term catalysis durability of CoP?

3. According to the AIMD analysis, manganese dioxide exhibits the strongest repulsion capability against Cl^- . How do manganese dioxide and phosphate ions synergistically influence the corrosion repellency performance? The author should provide additional clarification on this aspect.

4. In order to better illustrate the performance of the catalyst in practical applications, it is necessary to further clarify the iR correction process, determine whether the data is compensated, and avoid overcompensation of the iR potential drop.

5. The authors have provided multiple lines of evidence demonstrating that the cage-like nanoarray facilitates bubble release and thus enhances mechanical stability, which is beneficial. Corresponding bubble adhesion experiments may further strengthen the conclusions.

6. While $\text{MnO}_2@\text{CoPi}@\text{CoP}/\text{NF}$ demonstrates exceptional stability in a two-electrode device, the performance of $\text{MnO}_2@\text{CoPi}@\text{CoP}/\text{NF}$ at a larger size remains unexplored. Evaluating scaled-up configurations would significantly raise the comprehensiveness and impact of the work.

Reviewer #2

(Remarks to the Author)

This study proposes a new electrode design to address the challenges of seawater electrolysis. The electrode consists of two distinct layers: a Co-phosphate (Co-Pi) outer layer that repels Cl^- ions and a layer of MnO_2 nanoparticles that further filter out Cl^- while mitigating the chlorine evolution reaction and corrosion. Additionally, the authors highlight that a cage-like structure, formed by tip-connected nanowires, minimizes physical stress from oxygen bubble movement, thereby enhancing mechanical stability. Although a combination of in situ Raman, TOF-SIMS, XPS, and DFT calculations attempts to support these assertions, several major concerns remain. In particular, the merely simple comparison of catalyst lifespan appears to be highly misleading. Specific concerns are outlined below.

1. The authors present Figure 1 as the very first figure in the manuscript to compare the lifetimes of recently reported catalysts and highlight the notable performance of the catalyst in this study. However, we would like to note that this way of data presentation is completely misleading. A comparison of catalysts lifetime, without standardizing key parameters such as electrode conditions (including mass loading and surface area of the catalysts) and current-density test conditions, does not provide any scientifically objective evaluation. For example, Ref. 18 (Fan, R. et al. Ultrastable electrocatalytic seawater

splitting at ampere-level current density, Nat. Sustain. 7, 158–167 (2024)) in the manuscript already reports a 2800-h duration at 1.25 A/cm², whereas this study presents a 3000-h duration at 1.0 A/cm². Few researchers would consider this as evidence of superior performance. We believe that the current trend of 'lifetime races' in the field of OER catalysts is NOT the constructive direction for advancing relevant research.

2. Furthermore, the authors seem to have (intentionally?) overlooked several notable results from previous studies, which may lead to an overemphasis on the findings of the present study. For a more objective comparison, reports of catalyst lifetimes exceeding 10,000 h should be considered. For example, Adv. Mater. 36, 2411302 (2024) presents results for seawater electrolysis, Nature Catalysis 7, 944–952 (2024) reports 15,000 h at 1 A/cm² for alkaline electrolysis, and Nature 639, 360–367 (2025) demonstrates 10,000-h stability in intermittent alkaline seawater electrolysis.

3. One of the most puzzling assertions in this study is that a negatively charged phosphate layer can passivate the outer surface as the first line of defense by preferentially repelling Cl⁻ anions, as shown in Figure 2a. If this is the case, the same negatively charged phosphate layer should also repel OH⁻ anions, which would ultimately lead to a significant reduction in OER activity. Without clarifying this seemingly contradictory selective repulsion, few readers would fully understand the role of the outer layer.

4. The authors repeatedly refer to 'phosphorus-related species' or '(PO₄)₃⁻-related species' throughout the manuscript without precisely identifying the corresponding peaks in the Raman spectra. It is important to note that the presence of peaks associated with (PO₄)₃⁻-related species does not necessarily indicate the exclusive presence of (PO₄)₃⁻ at the surface. For charge neutrality, Co²⁺ must also be present. This appears to be the most significant misunderstanding.

5. Figure 4b shows that the Raman peaks for phosphates appear below 1.23 V for the Co-Pi@CoP/NF and MnO₂@Co-Pi@CoP/NF samples. Theoretically, the oxygen reduction reaction (ORR) should occur below 1.23 V. This raises the question of what drives the oxidation of phosphorus under these conditions.

6. We find it unlikely that the surfaces of CoP and even MnO₂ remain crystalline after prolonged OER operation. Therefore, the charge distributions in Figure 5h, which are derived from crystalline supercells, may be misleading.

7. The simultaneous coexistence of positive and negative pressures as a pair in Figure 6g seems highly unlikely. In particular, maintaining such a large local pressure difference (from positive to negative values) at the tip would be extremely challenging. Readers may question the reliability of how this pressure difference was obtained.

8. From pages 7 to 8, 'Figure 1' should have been labeled as 'Figure 2.' Additionally, in the section 'Catalytic Activity Improvements,' there are no main figures, as all related figures are included in the Supplementary Information. Furthermore, most paragraphs are excessively long. These editorial inconsistencies, including incorrect figure numbering, make it extremely difficult to follow the content of this study. We believe that careful manuscript preparation is the essential first step before publication.

Reviewer #3

(Remarks to the Author)

Li et al. discuss an MnO₂/Co-phosphate/CoP catalyst for seawater electrolysis which shows good stability over 3000h at industrial-level current densities. This stability is attributed to both the morphology and the layered composition of the material. Although the material is interesting, I don't immediately see the major advance made in the paper that would warrant publication in Nature Communications. I would therefore recommend publication in a more specialized journal. Below my detailed comments

1. The authors put quite some emphasis on the tip-connected morphology, which reduces the forces induced by bubbles during the OER. However, the evidence this plays a major role in the stability of the material is minimal. Experimentally, the authors only provide SEM images of Co-Pi (which allegedly does not have the tip-connected morphology, although this is not very clear to me) and MnO₂-CoPi before and after the OER. I cannot say I see much difference in morphology between any of the images. If the authors really want to argue that this factor is important (for which I have seen very little evidence in the literature), they should provide more solid evidence.

2. A key factor in seawater electrolysis is the selectivity for the OER vs. CER. The authors provide no direct measurement of this. Only some ClO⁻ concentration measurements are done after long-term measurements, but solutions after different electrolysis times with different samples and different current densities are compared. No direct detection of the Cl₂/O₂ ratio is provided.

3. The discussion of the catalyst structure that actually does the job is a bit limited. First, the authors do not discuss what happens to the cobalt species during electrolysis. Their post-OER XRD shows that the CoP peaks are lost (so their argument that CoP is important for providing conductivity does not seem validated). post-OER Co 2p XPS would be good to further confirm this. Secondly, the authors do not discuss the likely accumulation of contaminants such as Fe on the catalyst, which is very usual for 3d metal oxide catalysts during OER and greatly affects the activity.

4. For educational purposes, I feel it is important that the authors correctly represent the "business case" for direct seawater electrolysis. The authors claim that pure water electrolyzers have to rely on fresh water. This is not the case. Seawater desalination is a well-established industrial process that can produce pure water for electrolysis, contributing only ~2% to the

overall energy cost. The reason that direct seawater electrolysis could be of interest is that it reduces the footprint and CAPEX of the system, which might be interesting for e.g. offshore applications.

5. For the activity measurements, especially at these very high current densities, it is crucial to know what level of iR correction was applied. This is not reported with the data. However, the data in supp fig 5 looks over-corrected, particularly the red curve. This is also apparent from supp fig 6, where it can be seen that the apparent Tafel slope decreases at high current density, which is very unusual behavior. Later figures don't show this apparent iR correction and indeed look much more usual. In general, I am not sure whether a perfect iR correction is possible at very high current densities, given that the bubble behavior, local temperature, and pH gradients may drastically change the solution resistance. Therefore, for the discussion of the intrinsic overpotential of the catalyst, the authors should focus on the low-current density regime and only apply an iR correction here (and clearly state it). Other data should not be iR corrected.

6. Some description on how the TOF is calculated is necessary. Did the authors assume that any redox-active site is OER active?

7. The activation energies are calculated from data obtained at very high overpotential, where mass/charge transport resistance plays a huge role. Therefore, the obtained values may reflect mass transport rather than the OER barrier. It would be better to do the calculation with low overpotential data.

8. I think the authors may misinterpret their pH and cation dependence data. 3d metal oxides often become amorphous during the OER (the Co indeed seems to do that based on the XRD), after which cations can intercalate into the structure. This strongly modulates the OER performance of the material. When the pH is changed, the cation concentration is changed as well. Hence the difference between the different materials may primarily reflect the cation intercalation behavior.

9. Observing rare events like surface chlorination in AIMD is not likely. Therefore, I don't think the AIMD simulations carry much meaning for explaining the performance of the catalyst.

10. Are the durations reported in Fig. 3g the testing times, or the actual life spans of the catalysts? Of course, other materials may have also lasted for 3000h, but were just not tested that long. This creates an unfair comparison. A better figure of merit would be the overpotential increase per hour.

11. In the Mn 2p analysis in Fig. 5b, the authors focus on the energy difference between the 2p_{3/2} and 2p_{1/2} to analyse the chemical state of the Mn. However, this spin-orbit splitting is essentially independent of the oxidation state (in contrast to the splitting of the Mn 3s, which is not due to spin-orbit coupling). It would make more sense if the authors look at the position of the Mn 2p_{3/2} peak.

12. Fig 1 a-c are not very informative. Fig 1d is misleading, as it suggests that electrolysis in pure water + KOH has only been tested up to a few hundred hours. In fact this electrolyte has been used/tested in industrial alkaline electrolyzers over their full lifespan of ~8 years.

13. Fig 2 The images in b are recorded at a very different angle than c, making it difficult to assess if the morphology really changed. Arguably, the bottom image of b also show tip-connected arrays. Furthermore, for STEM-EDX analysis in f is would be good to add an overlay image of all the elements to highlight the core-shell structure.

14. It is unclear what is shown in fig 3f

15. Supp fig 1: please note what materials the PDFs belong to

16. Supp fig 9d: should the unit be V instead of mV?

Version 1:

Reviewer comments:

Reviewer #1

(Remarks to the Author)

This manuscript is recommended for publication now.

Reviewer #2

(Remarks to the Author)

While we previously expressed concern about the very limited value of "lifetime races" in the OER catalyst field, the authors continue to assert that reporting long-term stability remains meaningful. If this is the case, Figure 1 in the main text should comprehensively include representative and high-performance literature data for a fair and informative comparison. Although

some of this information is provided in Supplementary Table 1, the current form of Figure 1 remains unrevised and thus misleading. To improve transparency and ensure balanced interpretation, key prior results should be accurately and visibly plotted in a “current density vs. lifetime” format in Figure 1, without any intentional omission (Nat. Commun. 15, 2481 (2024) for 3000 h at 0.5 A/cm²; Nat. Sustain. 7, 158–167 (2024)) for a 2800-h duration at 1.25 A/cm²; Adv. Mater. 36, 2411302 (2024) for 10000 h at 0.4 A/cm²; Adv. Mater. 37, 2415421 (2025) for 2500 h at 1 A/cm²; and Nature 639, 360–367 (2025) for 10,000-h stability in intermittent seawater electrolysis).

Reviewer #3

(Remarks to the Author)

The authors have adequately addressed my comments. I believe that the manuscript is suitable for publication in its present form.

Point-by-Point Responses to Reviewers' Comments

We express our sincere gratitude to the editor and all reviewers for their invaluable feedback, which we have utilized to enhance the quality of our manuscript (NCOMMS-25-17688). The reviewer comments are presented in *italic* and **bold** font, while our responses, including the incorporation of additional figures, tables, descriptions, and other elements, are highlighted in **blue** text.

Point-by-point response to the reviewers #1

Reviewer #1 (Remarks to the Author): The authors pursue integrating a triple defense at the nanoscale to enhance chemical and mechanical lifespans of anode catalysts, which is impressive. In fact, it is still rare to simultaneously optimize both the mechanical and chemical stability of an electrode in a highly corrosive environment such as seawater. This makes the findings of this work interesting and of great universal significance. In situ and ex situ data nicely visualize and reveal the triple defense mechanisms, indeed ensuring the reliability. Therefore, this work represents a solid contribution to the field and should be accepted with minor revisions.

General Response: We appreciate the reviewer for the professional and constructive comments. According to the suggestions, we have made the corrections and the point-by-point responses are presented below.

Comment 1: This paper points out that Co-Pi can promote the generation of active sites, which is an important point for understanding the improvement of performance. A better characterization of the Co-Pi coating depth on CoP is essential, necessitating transmission electron microscope images to reveal the structure.

Response 1: We appreciate this comment and do understand your concerns. To ensure relative objectivity, we conducted detailed observations of the catalyst morphology at multiple locations. **Fig. R1** or **Supplementary Fig. 4** presents electron microscopy data from different regions of the sample, aiming to comprehensively reveal the thickness of the surface Co-Pi layer. As shown in **Fig. R1** or **Supplementary Fig. 4**, the amorphous Co-Pi layer exhibits a distinct interface with the crystalline CoP and MnO₂ phases.

Measurements taken from various regions indicate that the average thickness is approximately 9 nm.

Fig. R1 HRTEM images of MnO₂@Co-Pi@CoP at two different areas. Scale bar: 20 nm.

Comment 2: *From the results presented in Figure 3, it is evident that both manganese dioxide and phosphate ions contribute to the enhanced stability of cobalt phosphide under the current density of 2 A cm⁻². Specifically, what is the quantitative contribution of manganese dioxide to the stability improvement? Additionally, can manganese dioxide alone significantly enhance the long-term catalysis durability of CoP?*

Response 2: We appreciate the valuable suggestion. To quantitatively evaluate the contribution of manganese dioxide to the enhanced stability, we synthesized γ -MnO₂ and CoP powders and prepared inks by mixing them in various mass ratios (0:1, 1:3, 1:2, 1:1, 2:1, and 3:1). These inks were drop-cast onto acid-treated Ni foam (NF) and subjected to stability tests using a LAND CT2001A workstation. As shown in **Fig. R2**, increasing the amount of MnO₂ improved the stability of the catalyst. However, when the ratio reached 1:1, a decline in stability was observed. This may be attributed to the insufficient amount of catalytically active species, resulting in a higher overpotential. The elevated overpotential could lead to unavoidable exposure of active sites to chloride ions, causing leaching of the active components. Moreover, excessive voltage may compromise the structural integrity of the material. Therefore, an appropriate MnO₂/CoP ratio not only helps reduce the overpotential but also repels chloride ions

(Cl⁻), enhancing corrosion resistance. Additionally, we tested a catalyst in which MnO₂ was the sole protective component (MnO₂@CoO_x/NF) (Fig. R3 or Supplementary Fig. 17). Even under a high current density of 2 A cm⁻² in alkaline seawater, this catalyst remained stable for 300 h, demonstrating that MnO₂ alone can provide excellent anti-corrosion performance.

Fig. R2 Lifespans of different MnO₂/CoP ratios: (a) 0:1, (b) 1:3, (c) 1:2, (d) 1:1, (e) 2:1, and (f) 3:1 in alkaline seawater.

Fig. R3 Lifespans of MnO₂@CoO_x/NF at 2 A cm⁻² in alkaline seawater.

Comment 3: According to the AIMD analysis, manganese dioxide exhibits the strongest repulsion capability against Cl⁻. How do manganese dioxide and phosphate ions synergistically influence the corrosion repellency performance? The author should provide additional clarification on this aspect.

Response 3: We appreciate this comment. We have performed AIMD simulation for phosphate ions-modified manganese dioxide (PO₄³⁻@γ-MnO₂). As depicted in **Supplementary Fig. 26**, γ-MnO₂ (100) and PO₄³⁻@γ-MnO₂ (100) both turn out to have higher coverage of oxygen-containing intermediates, including *O, *OH, and *H₂O, while fewer oxygen-containing intermediates are generated on CoOOH (001), CoP (011), and Co-Pi (001), thus indicating γ-MnO₂ (100) and PO₄³⁻@γ-MnO₂ (100) possess strong adsorption of these species. As a result of different coverages of the oxygen-containing intermediates, PO₄³⁻@γ-MnO₂ (100) displays significant resistance to Cl⁻ than other materials. As shown in **Fig. 3f**, PO₄³⁻@γ-MnO₂ (100) has the longest average Cl⁻-to-surface distance of 8.16 Å (2.32 Å for CoP (011), 2.18 Å for Co-Pi (001), 3.59 Å for CoOOH (001), and 5.48 Å for γ-MnO₂ (100), respectively), demonstrating that the PO₄³⁻@γ-MnO₂ (100) has the strongest resistance to Cl⁻. It should be noted that Cl⁻ anions are adsorbed by the active sites of all the materials except γ-MnO₂ (100) and PO₄³⁻@γ-MnO₂ (100) (**Supplementary Fig. 26**). In addition, although *O species exist on γ-MnO₂ (100) and PO₄³⁻@γ-MnO₂ (100), no *OCl species can be found during the AIMD process, further demonstrating the excellent resistance of γ-MnO₂ to Cl⁻.

Comment 4: In order to better illustrate the performance of the catalyst in practical applications, it is necessary to further clarify the iR correction process, determine whether the data is compensated, and avoid overcompensation of the iR potential drop.

Response 4: We appreciate reviewer's highly specialized comment. The iR-compensated potential was obtained after correcting the solution resistance measured according to the equation: $E_{\text{corr}} = E - iR$, where E is the original potential, R is the solution resistance, i is the corresponding current, and E_{corr} is the iR-compensated potential. LSV curves in **Supplementary Fig. 6** and **24** were compensated with 85% iR. Other electrochemical measurements, such as stability, activation energy, etc., were conducted without iR compensation and just converted to the standard electrode potential.

Comment 5: The authors have provided multiple lines of evidence demonstrating that the cage-like nanoarray facilitates bubble release and thus enhances mechanical stability, which is beneficial. Corresponding bubble adhesion experiments may further strengthen the conclusions.

Response 5: We sincerely thank this reviewer for raising this insightful comment. We believe that the unique nanocage structure facilitates the release of O₂ bubbles and mitigates the mechanical impact caused by bubble transport and detachment. To support this, we conducted bubble adhesion force measurements on different samples, as shown in **Fig. R4** or **Supplementary Fig. 35**. Among them, MnO₂@Co-Pi@CoP/NF exhibits the lowest bubble adhesion force (~6.2 μN) due to its distinctive cage-like nanoarray structure, compared to CoP/NF (~10.6 μN) and Co-Pi@CoP/NF (~10.4 μN), which promote more efficient bubble detachment. This effect is particularly beneficial under high current densities, where timely bubble release helps maintain the exposure of active sites to the electrolyte, thereby minimizing activity degradation.

Fig. R4 Adhesive forces curves of (a) CoP/NF, (b) Co-Pi@CoP/NF, and (c) MnO₂@Co-Pi@CoP/NF. (d) The corresponding adhesive forces data.

Comment 6: While MnO₂@CoPi@CoP/NF demonstrates exceptional stability in a two-electrode device, the performance of MnO₂@CoPi@CoP/NF at a larger size remains unexplored. Evaluating scaled-up configurations would significantly raise the comprehensiveness and impact of the work.

Response 6: We sincerely thank this reviewer for raising this insightful comment. We fabricated electrodes with an area of 9 cm² and evaluated them in an anion exchange membrane electrolyzer (AEME). Despite the increase in electrode area from 1 to 9 cm², the performance of the resulting AEME does not show significant degradation (Fig. R5), indicating that MnO₂@Co-Pi@CoP/NF is well-suited for large-area fabrication while maintaining excellent electrochemical activity.

Fig. R5 Polarization curves recorded for Pt/C/NF || MnO₂@Co-Pi@CoP/NF with different electrode areas.

Point-by-point response to the reviewers #2

Reviewer #2 (Remarks to the Author): This study proposes a new electrode design to address the challenges of seawater electrolysis. The electrode consists of two distinct layers: a Co-phosphate (Co-Pi) outer layer that repels Cl^- ions and a layer of MnO_2 nanoparticles that further filter out Cl^- while mitigating the chlorine evolution reaction and corrosion. Additionally, the authors highlight that a cage-like structure, formed by tip-connected nanowires, minimizes physical stress from oxygen bubble movement, thereby enhancing mechanical stability. Although a combination of in situ Raman, TOF-SIMS, XPS, and DFT calculations attempts to support these assertions, several major concerns remain. In particular, the merely simple comparison of catalyst lifespan appears to be highly misleading. Specific concerns are outlined below.

General Response: Great thanks for your supportive feedback on our manuscript. We appreciate your time and effort in reviewing our work and providing professional and invaluable suggestions. We have followed your suggestions to further improve the quality of our manuscript. The detailed corrections are listed below.

Comment 1: The authors present Figure 1 as the very first figure in the manuscript to compare the lifetimes of recently reported catalysts and highlight the notable performance of the catalyst in this study. However, we would like to note that this way of data presentation is completely misleading. A comparison of catalysts lifetime, without standardizing key parameters such as electrode conditions (including mass loading and surface area of the catalysts) and current-density test conditions, does not provide any scientifically objective evaluation. For example, Ref. 18 (Fan, R. et al. Ultrastable electrocatalytic seawater splitting at ampere-level current density, Nat. Sustain. 7, 158–167 (2024)) in the manuscript already reports a 2800-h duration at 1.25 A/cm^2 , whereas this study presents a 3000-h duration at 1.0 A/cm^2 . Few researchers would consider this as evidence of superior performance. We believe that the current trend of ‘lifetime races’ in the field of OER catalysts is NOT the constructive direction for advancing relevant research.

Response 1: Thanks for raising this valuable comment. We sincerely appreciate the reviewer’s insightful comment regarding the scientific objectivity of lifetime comparisons among different catalysts. We actually agree that lifetime, when presented without standardized testing conditions, may lead to ambiguous or even misleading interpretations. In response, we have carefully revised the manuscript by including a more detailed and transparent lifetime comparison table (**Table R1** or **Supplementary Table 1**), summarizing key testing parameters such as electrolyte, current density (j), mass loading, surface area, temperature, and test system. This provides a more contextual and objective basis for evaluating the long-term performance of our catalyst relative to other state-of-the-art systems. While we acknowledge the reviewer’s concern about the ongoing “lifetime race” in the OER field, we would also like to emphasize that operational durability remains a crucial criterion for practical water electrolysis, especially under high current densities relevant to industrial applications. Therefore, we believe it is still meaningful to report long-term stability data—provided it is done with sufficient contextual information—as it allows readers to judge performance comprehensively. We have clarified these points in the revised text and hope that the improved data presentation addresses the reviewer’s concern.

Table R1. Research efforts to enhance the OER electrode lifespan in the past few years. Details: electrolyte, j , time (t), mass loading (M), surface area (s), temperature (T), and test system (2 or 3 electrodes).

Catalyst	Electrolyte	j (mA cm ⁻²)	t (h)	M (mg cm ⁻²)	s (cm ⁻²)	T (°C)	System	Ref.
MnO ₂ @Co-Pi@CoP/NF	1 M KOH + seawater	1000	3000	~2.7	0.25	~25 °C	3	This work
Fe-NiSOH/NF	1 M KOH + seawater	500	900	-	-	-	3	Energy Environ. Sci. 15 , 4647–4658 (2022)

BZ-NiFe-LDH/NF	1 M KOH + seawater	500	100	-	0.25	~25 °C	3	Nano Res. Energy 1 , e9120028 (2022)
(NiFe) ₂ C ₂ O ₄ /NF	1 M KOH + seawater	1000	600	-	0.25	~25 °C	3	Angew. Chem. Int. Ed. 63 , e202316522 (2024)
CrO ₄ ²⁻ -NiFe LDH/Cr ₂ O ₃ /NF	1 M KOH + seawater	1000	1000	~2.0	0.25	~25 °C	3	Nat. Commun. 15 , 6624 (2024)
NiMoS _x @NiFe- LDH/NF	1 M KOH + seawater	500	500	-	0.25	-	3	Inorg. Chem. Front. 10 , 2766– 2775 (2023)
NiIr-LDH/NF	1 M KOH + seawater	500	650	1.0	1	-	2	J. Am. Chem. Soc. 144 , 9254–9263 (2022)
RuMoNi/NF	1 M KOH + seawater	500	3000	3.6	1	20 ± 2 °C	3	Nat. Commun. 14 , 3607 (2023)
MoO ₃ @CoO/CC	1 M KOH + seawater	600	1000	-	-	~25 °C	3	Nat. Commun. 15 , 2481 (2024)
Ag/NiFe LDH/NF	1 M KOH + seawater	1000	1000	-	-	-	3	Nano Energy 98 , 107212 (2022)
CoFePBA/Co ₂ P/N F	1 M KOH + seawater	1000	1000	-	-	~25 °C	3	Angew. Chem. Int. Ed. 62 , e202309882 (2023)
NiFe-LDH@Ag/NF	1 M KOH + seawater	400	2500	-	1	~25 °C	2	Adv. Mater. 36 , 2306062 (2024)
CoFe- Ci@GQDs/NF	1 M KOH + 0.5 M NaCl	1250	2800	-	-	~25 °C	3	Nat. Sustain. 7 , 158–167 (2024)

Os- Ni ₄ Mo/MoO ₂ /NF	1 M KOH + seawater	500	2500	-	-	~25 °C	3	Adv. Mater. 36 , 2408982 (2024)
Ni(OH) ₂ -PA- Fe/NF	1 M KOH + seawater	1000	1200	-	-	-	3	Green Chem. 27 , 464–472 (2025)
NFB-LDH/Ni mesh	1 M NaOH + seawater + 0.05 M Na ₂ SO ₄	400	10000	-	1	~25 °C	2	Adv. Mater. 36 , 2411302 (2024)
LiFePO ₄ (Ni(OH) ₂)/ L-LFP)	1 M KOH + seawater	100	600	-	-	-	3	Angew. Chem. Int. Ed. 63 , e202410396 (2024)
Ni ₃ FeN@PO ₄ ³⁻ /N F	1 M KOH + seawater	1000	2500	-	-	~25 °C	3	Adv. Mater. 37 , 2415421 (2025)
B, Fe-CoP	1 M KOH + seawater	100	200	-	-	-	3	Adv. Funct. Mater. 34 , 2402264 (2024)
Ni-BDC/NH ₂ -MIL- 88B(Fe)	1 M KOH + seawater	360	28	-	-	~25 °C	3	Adv. Funct. Mater. 34 , 2314611 (2024)
NiFe- MOF@NiS/NF	1 M KOH + seawater	100	600	-	-	-	3	Chin. J. Catal. 61 , 192–204 (2024)
(FeCoNiMnAl) ₃ O ₄ /NF	1 M KOH + seawater	500	50	-	-	-	3	Appl. Catal. B Environ. Energy 349 , 123875 (2024)
CP/NF	1 M KOH	200	40	5.2	1	-	3	Chem. Eng. J. 408 , 127331 (2021)

NiFe-P _{Zn} @PNTA	1 M KOH	100	360	-	1	~25 °C	3	Adv. Mater. 35 , 2209500 (2023)
C-Ni _{1-x} O/3DPNi	1 M KOH	850	16	-	-	80 °C	2	Adv. Energy Mater. 10 , 2002955 (2020) ACS Appl. Energy Mater. 3 , 9769– 9784 (2020)
CPNCP/NF	1 M KOH	200	17	5.2	1	-	3	Mater. 3 , 9769– 9784 (2020)

Comment 2: Furthermore, the authors seem to have (intentionally?) overlooked several notable results from previous studies, which may lead to an overemphasis on the findings of the present study. For a more objective comparison, reports of catalyst lifetimes exceeding 10,000 h should be considered. For example, *Adv. Mater.* **36**, 2411302 (2024) presents results for seawater electrolysis, *Nature Catalysis* **7**, 944–952 (2024) reports 15,000 h at 1 A/cm² for alkaline electrolysis, and *Nature* **639**, 360–367 (2025) demonstrates 10,000-h stability in intermittent alkaline seawater electrolysis.

Response 2: We sincerely thank the reviewer for this insightful comment. We acknowledge that several important recent studies reporting exceptional catalyst lifetimes were not included in the initial version of the manuscript. To provide a more comprehensive and objective comparison, we have now incorporated the following representative works into the revised manuscript: *Adv. Mater.* **36**, 2411302 (2024), *Nat. Catal.* **7**, 944–952 (2024). These studies have been cited and briefly discussed in the revised version to better contextualize the durability performance of our catalyst. In addition, we have included a detailed comparison table (**Table R1** or **Supplementary Table 1**) summarizing the catalyst lifetimes from a wide range of recent reports, including the works mentioned above. This table highlights not only the stability data but also key testing parameters such as electrolyte composition, *j*, and electrode configuration, thereby offering a more objective and transparent benchmark for

evaluating long-term performance. However, the report (*Nature* **639**, 360–367 (2025)) aims to study the catalyst for alkaline seawater reduction, which is not directly comparable to our work. Because the reactions and challenges involved are different, we did not compare them in this work.

Comment 3: One of the most puzzling assertions in this study is that a negatively charged phosphate layer can passivate the outer surface as the first line of defense by preferentially repelling Cl⁻ anions, as shown in Figure 2a. If this is the case, the same negatively charged phosphate layer should also repel OH⁻ anions, which would ultimately lead to a significant reduction in OER activity. Without clarifying this seemingly contradictory selective repulsion, few readers would fully understand the role of the outer layer.

Response 3: We fully understand your doubts about the selective repulsion of the negatively charged phosphate layer. This is indeed an aspect that we did not explain sufficiently in the paper, and we sincerely apologize for this. In fact, with the current accuracy of experimental techniques and theoretical simulations, it is not possible to definitively determine whether the phosphate layer exclusively repels Cl⁻. Nevertheless, we provide the most plausible explanation based on the available evidence. We also agree that, theoretically, a negatively charged phosphate layer should repel both OH⁻ and Cl⁻ anions, which might lead to a significant reduction in the OER activity. However, this selective repulsion is actually the result of the synergy of multiple factors, and we will explain it in detail as follows:

1. We have performed AIMD simulation for phosphate ions-modified manganese dioxide (PO₄³⁻@γ-MnO₂). As depicted in **Supplementary Fig. 26**, γ-MnO₂ (100) and PO₄³⁻@γ-MnO₂ (100) both turn out to have higher coverage of oxygen-containing intermediates, including *O, *OH, and *H₂O, while fewer oxygen-containing intermediates are generated on CoOOH (001), CoP (011), and Co-Pi (001), thus indicating γ-MnO₂ (100) and PO₄³⁻@γ-MnO₂ (100) possess strong adsorption of these species. As a result of different coverages of the oxygen-containing intermediates, PO₄³⁻@γ-MnO₂ (100) displays significant resistance to Cl⁻ than

other materials. As shown in **Fig. 3f**, $\text{PO}_4^{3-}@ \gamma\text{-MnO}_2$ (100) has the longest average Cl^- -to-surface distance of 8.16 Å (2.32 Å for CoP (011), 2.18 Å for Co-Pi (001), 3.59 Å for CoOOH (001), and 5.48 Å for $\gamma\text{-MnO}_2$ (100), respectively), demonstrating that the $\text{PO}_4^{3-}@ \gamma\text{-MnO}_2$ (100) exhibits the strongest resistance to Cl^- .

2. Although it is impossible to determine whether only Cl^- is excluded, manganese oxide will preferentially adsorb OH^- . Moreover, MnO_2 exhibits strong Lewis acid characteristics, which can enrich OH^- evidenced by the methanol oxidation experiment (**Supplementary Fig. 24**), without being affected by electrostatic repulsion, while Cl^- will be repelled.
3. Difference in ion size. The ionic radius of Cl^- is larger than that of OH^- . The negatively charged phosphate layer has a certain pore structure. The size of these pores allows OH^- to pass through, but acts as a physical barrier to the larger Cl^- (*Surf. Coat. Technol.* **207**, 50–65 (2012)). It is like a sieve that can screen out ions of appropriate size.
4. Difference in ion hydration layers. Both Cl^- and OH^- form hydration layers in aqueous solutions, but differences in ion hydration may lead to different affinity outcomes.
5. In alkaline seawater, the concentration of OH^- is higher than that of Cl^- . Additionally, strong hydrogen bonding between oxyhydroxides and OH^- prevents Cl^- from experiencing electrostatic repulsion, as reported in *ACS Catal.* **13**, 15360–15374 (2023). This phenomenon leads to increased repulsion of Cl^- and absorption of OH^- .

Comment 4: The authors repeatedly refer to ‘phosphorus-related species’ or ‘ $(\text{PO}_4)^{3-}$ -related species’ throughout the manuscript without precisely identifying the corresponding peaks in the Raman spectra. It is important to note that the presence of peaks associated with $(\text{PO}_4)^{3-}$ -related species does not necessarily indicate the exclusive presence of $(\text{PO}_4)^{3-}$ at the surface. For charge neutrality, Co^{2+} must also be present. This appears to be the most significant misunderstanding.

Response 4: Thank you for the reminder. To address this problem, we re-analyzed the Raman spectral data carefully, clearly identified the peaks (in situ generated PO_4^{3-}), and marked them clearly in the figure. Additionally, we also supplemented relevant references to support our judgment on the assignment of Raman spectral peaks (*Angew. Chem. Int. Ed.* **63**, e202410396 (2024), *Adv. Mater.* **37**, 2415421 (2025), *Angew. Chem. Int. Ed.* **62**, e202309882 (2023), *ACS Nano* **19**, 1530–1546 (2025), *Adv. Funct. Mater.* **35**, 2417211 (2025)), making the content of the manuscript more scientific and rigorous. As shown in **Fig. 4b**, the peaks at 977.5 and 1059.6 cm^{-1} correspond to the symmetric stretching vibration ($\nu_{\text{sym, P-O}}$) and the asymmetric stretching vibration ($\nu_{\text{asym, P-O}}$) of protonated PO_4^{3-} (*J. Am. Chem. Soc.* **141**, 15891–15900 (2019)), respectively. It indicates that the Co-Pi layers in both Co-Pi@CoP/NF and MnO_2 @Co-Pi@CoP/NF contain two different types of phosphate species, with protonated PO_4^{3-} -related species being the dominant form based on peak intensity.

It is noted that PO_4^{3-} detected during OER originates in situ from the structural transformation of the pre-catalyst phases (Co-Pi and CoP). Under anodic potentials, these phases undergo oxidation: CoP releases $\text{P}^{\delta-}$ (oxidized to PO_4^{3-}) while Co is oxidized to higher valence states ($\text{Co}^{3+}/\text{Co}^{4+}$), forming active Co oxyhydroxides (CoOOH), which are primarily stabilized by reaction intermediates. We do not deny the existence of Co ions, but the surface does not necessarily have to remain charge neutrality during electrochemical reactions. Some PO_4^{3-} -related species may adsorb on the surface to interfere with the adsorption of Cl^- (*Angew. Chem. Int. Ed.* **60**, 22740 (2021), *Nat. Commun.* **14**, 3607 (2023), *Adv. Mater.* **36**, 2311322 (2024), *Angew. Chem. Int. Ed.* **62**, e20230988 (2023)).

Comment 5: *Figure 4b shows that the Raman peaks for phosphates appear below 1.23 V for the Co-Pi@CoP/NF and MnO_2 @Co-Pi@CoP/NF samples. Theoretically, the oxygen reduction reaction (ORR) should occur below 1.23 V. This raises the question of what drives the oxidation of phosphorus under these conditions.*

Response 5: Thanks very much for raising this concern. The phosphates existed before the electrochemical test. The Raman peaks for phosphates appear below 1.23 V for

the Co-Pi@CoP/NF are caused by an anodic electrochemical treatment, and the Raman peaks for phosphates appear below 1.23 V for the MnO₂@Co-Pi@CoP/NF are caused by spontaneous redox.

In the case of Co-Pi@CoP/NF, the preparation involves the polarization of CoP/NF in 0.5 M Na₃PO₄. During this process, low-valent Co^{δ+} and P^{δ-} in CoP/NF are oxidized to high-valent Co species and phosphate ions, eventually forming a Co-Pi layer on the surface of CoP/NF. As a result, phosphate-related peaks can be observed in the Raman spectra even at potentials below 1.23 V. Similarly, when CoP/NF is immersed in 0.5 M KMnO₄, a spontaneous redox reaction occurs between MnO₄⁻ and CoP. Mn⁷⁺ is reduced to Mn⁴⁺, forming MnO₂, while Co^{δ+} and P^{δ-} are oxidized to Co³⁺/Co⁴⁺ and PO₄³⁻, respectively, resulting in the formation of a composite surface layer containing MnO₂ and Co-Pi. Therefore, characteristic peaks associated with phosphate species can also be observed in the Raman spectra at the initial state. In summary, the presence of PO₄³⁻-related peaks below 1.23 V does not indicate the occurrence of phosphate formation during electrochemical oxidation but reflects the phosphate species already formed during the preparation of the catalyst.

Comment 6: We find it unlikely that the surfaces of CoP and even MnO₂ remain crystalline after prolonged OER operation. Therefore, the charge distributions in Figure 5h, which are derived from crystalline supercells, may be misleading.

Response 6: Thank you for your critical insight into the theoretical calculations. We acknowledge the concern that material surfaces like MnO₂ may lose crystallinity during prolonged OER. Here is our detailed response:

1. We agree that electrochemical corrosion, ion insertion/extraction, or mechanical stress during OER may induce amorphous surface reconstruction. The charge distribution in **Fig. 5h**, derived from pristine crystalline supercells, does not directly reflect the reconstructed surface structure, which is a limitation of the current model.
2. In situ XRD, Mn 2p and Mn 3s XPS data, and HRTEM images should support the existence of crystalline MnO₂ species.

3. CoP itself is not located on the surface but in the innermost part of the catalyst, where it undergoes a relatively slow conversion into hydroxide/oxide species. This slower transformation may help maintain more crystalline phases at the surface.
4. Surface reconstruction or amorphization may occur during long-term OER operation, and we fully acknowledge that the actual catalyst surfaces could differ from the idealized crystalline structures used in our calculations. However, it is widely recognized in computational studies that using crystalline surface models remains a valid and informative approach for evaluating key material properties such as ion adsorption. These models reflect the intrinsic electronic structures and coordination environments of the materials, which fundamentally govern the adsorption behavior of Cl^- .
5. DFT calculations are not intended to reproduce the exact atomic structure of the catalyst under operando conditions, but rather to provide a comparative understanding of the Cl^- adsorption affinity across different materials. This approach allows for meaningful interpretation of material-dependent trends. Notably, the computational results are consistent with our experimental observations, reinforcing the relevance and reliability of the theoretical insights derived from these models.

Comment 7: The simultaneous coexistence of positive and negative pressures as a pair in Figure 6g seems highly unlikely. In particular, maintaining such a large local pressure difference (from positive to negative values) at the tip would be extremely challenging. Readers may question the reliability of how this pressure difference was obtained.

Response 7: We thank the reviewer for raising this comment and do understand your concerns. Firstly, velocity is a vector quantity with directionality. Therefore, the pressure distribution calculated from the velocity field may also exhibit directional variations. In computational fluid dynamics, negative pressure values do not necessarily indicate physical “negative pressure,” but rather reflect locally low-pressure regions relative to a reference pressure, or numerical characteristics resulting

from changes in the flow direction. In fluid flow, pressure differences are the fundamental driving force. Fluid naturally tends to move from regions of high pressure to regions of low pressure. For example, in the contraction section of a pipe, as the flow velocity increases, the pressure decreases. The upstream region may exhibit positive pressure, while the downstream region may show a drop in pressure—potentially even negative values—due to the acceleration of the fluid. It illustrates Bernoulli's principle, where an increase in flow velocity leads to a decrease in pressure. In addition, in complex flow fields, such as the wake region behind objects, local vortex zones, flow separation, and recirculation can result in local low-pressure regions, where the pressure may fall below the surrounding ambient pressure and appear negative. In the present model with a hollow conical structure, local vortices are likely to form near the top and central regions, leading to directional variations in the pressure field and the appearance of negative values. Such pressure behavior is a common and physically reasonable phenomenon in systems with geometric discontinuities or unsteady flow.

Comment 8: From pages 7 to 8, 'Figure 1' should have been labeled as 'Figure 2.' Additionally, in the section 'Catalytic Activity Improvements,' there are no main figures, as all related figures are included in the Supplementary Information. Furthermore, most paragraphs are excessively long. These editorial inconsistencies, including incorrect figure numbering, make it extremely difficult to follow the content of this study. We believe that careful manuscript preparation is the essential first step before publication.

Response 8: Thanks for the kind reminder. As suggested, we have corrected the errors in the corresponding sections and streamlined the content in the "Catalytic activity improvements" section to facilitate better readability and understanding for the readers.

Point-by-point response to the reviewers #3

Reviewer #3 (Remarks to the Author): *Li et al. discuss an MnO₂/Co-phosphate/CoP catalyst for seawater electrolysis, which shows good stability over 3000h at industrial-level current densities. This stability is attributed to both the morphology and the layered composition of the material. Although the material is interesting, I don't immediately see the major advance made in the paper that would warrant publication in Nature Communications. I would therefore recommend publication in a more specialized journal. Below my detailed comments.*

General Response: We sincerely appreciate the reviewer for the insightful comments on our work. We acknowledge that your comments are important for us to improve the quality of our work. We have studied all comments carefully and have made corrections, which we sincerely hope will meet with approval.

Comment 1: *The authors put quite some emphasis on the tip-connected morphology, which reduces the forces induced by bubbles during the OER. However, the evidence this plays a major role in the stability of the material is minimal. Experimentally, the authors only provide SEM images of Co-Pi (which allegedly does not have the tip-connected morphology, although this is not very clear to me) and MnO₂-CoPi before and after the OER. I cannot say I see much difference in morphology between any of the images. If the authors really want to argue that this factor is important (for which I have seen very little evidence in the literature), they should provide more solid evidence.*

Response 1: We appreciate the reviewer's valuable comments regarding the role of the tip-connected morphology in enhancing the catalyst stability. In response, we have taken several steps to strengthen our argument and provide more convincing evidence: First, to better evaluate the morphological evolution of the catalyst during OER, we have supplemented additional SEM images of Co-Pi@CoP/NF and MnO₂@Co-Pi@CoP/NF after the long-term electrolysis (**Fig. R1** or **Fig. 3d,e**). **Fig. R1a** or **Fig. 3d** demonstrates that MnO₂@Co-Pi@CoP/NF maintains excellent structural integrity after 10-h continuous electrolysis at 0.5 A cm⁻² in KOH solution. In contrast, Co-Pi@CoP/NF

electrode exhibits significant structural degradation, characterized by broken structures and severe agglomeration. Building on this confirmation of enhanced mechanical robustness, we further compared the anode morphologies after eASO under high current density (j). As shown in **Fig. R1b** or **Fig. 3e**, the counterpart electrode experiences obvious corrosion and damage from gas bubble impact after just 5 h of electrolysis, whereas $\text{MnO}_2@\text{Co-Pi}@CoP/NF$ retains its nanocage-like morphology without visible changes. Long-term electrolysis aggravates the detachment of the control sample, with intensified corrosion and increased nanowire aggregation. In contrast, $\text{MnO}_2@\text{Co-Pi}@CoP/NF$ electrode preserves its original tip-connected nanowire architecture even after extended eASO, which suggests that both the mechanical and chemical stability are improved. Second, we have performed bubble adhesion force measurements for both samples using a high-precision microforce tester (**Fig. R2**). The results indicate that the $\text{MnO}_2@\text{Co-Pi}@CoP/NF$ exhibits significantly lower bubble adhesion force ($\sim 6.2 \mu\text{N}$) compared to $\text{Co-Pi}@CoP/NF$ ($\sim 10.4 \mu\text{N}$), which implies more efficient bubble detachment, thereby mitigating mechanical stress accumulation on the surface. Together, post-reaction SEM images as well as our simulation study (**Fig. 6**) provide direct experimental and theoretical evidence to double confirm the role of the tip-connected morphology in stabilizing the catalyst under alkaline seawater oxidation, with bubble adhesion force further quantifying the effect of the tip-connected structure.

Fig. R1 (a) SEM images after 10-h OER tests. (b) SEM images after eASO tests at ultrahigh j . Scale bar: $1 \mu\text{m}$.

Fig. R2 Adhesive forces data of Co-Pi@CoP/NF and MnO₂@Co-Pi@CoP/NF.

Comment 2: A key factor in seawater electrolysis is the selectivity for the OER vs. CER. The authors provide no direct measurement of this. Only some ClO⁻ concentration measurements are done after long-term measurements, but solutions after different electrolysis times with different samples and different current densities are compared. No direct detection of the Cl₂/O₂ ratio is provided.

Response 2: We sincerely thank the reviewer for the valuable comment. The selectivity between the oxygen evolution reaction and the chlorine evolution reaction is indeed a critical issue in seawater electrolysis. In this study, our primary focus was to evaluate the structural stability and corrosion resistance of the catalyst under practical alkaline seawater conditions. Therefore, we initially employed the concentration of ClO⁻ after long-term electrolysis as an indirect indicator to assess CER activity. However, we acknowledge that this method has limitations, especially when comparing different samples and j , which could affect the accuracy of selectivity evaluation. To address this issue, we have supplemented the data with ClO⁻ concentrations measured under the same electrolysis time and j . As shown in **Fig. R3** or **Supplementary Fig. 19**, UV-vis spectra of ClO⁻ generated from CoP/NF, Co-Pi@CoP/NF, and MnO₂@Co-Pi@CoP/NF after 50 and 100-h electrolysis at 1 A cm⁻² reveal a clear trend: MnO₂@Co-Pi@CoP/NF produced the least amount of ClO⁻, followed by Co-Pi@CoP/NF, while CoP/NF produced the most. In addition, we agree that direct detection of Cl₂/O₂ product ratios would provide a more accurate assessment of reaction selectivity. To this end, we

attempted to use differential electrochemical mass spectrometry to detect Cl_2 generation during electrolysis. However, there is no Cl_2 signal that can be observed (**Fig. R4**), likely due to the rapid disproportionation of Cl_2 into ClO^- in alkaline media, making it difficult to detect Cl_2 directly. As an alternative, we measured the Faradaic efficiency of O_2 evolution for $\text{MnO}_2@\text{Co-Pi}@\text{CoP}/\text{NF}$, which is found to be as high as 99% (**Supplementary Fig. 22**), further confirming the high OER selectivity of our catalyst.

Fig. R3 UV-vis absorption spectra of electrolytes from (a) CoP/NF, (b) Co-Pi@CoP/NF, and (c) $\text{MnO}_2@\text{Co-Pi}@\text{CoP}/\text{NF}$ after tests at 1 A cm^{-2} .

Fig. R4 Online mass spectrometry spectra of (a) CoP/NF, (b) Co-Pi@CoP/NF, and (c) MnO₂@Co-Pi@CoP/NF.

Comment 3: *The discussion of the catalyst structure that actually does the job is a bit limited. First, the authors do not discuss what happens to the cobalt species during electrolysis. Their post-OER XRD shows that the CoP peaks are lost (so their argument that CoP is important for providing conductivity does not seem validated). post-OER Co 2p XPS would be good to further confirm this. Secondly, the authors do not discuss the likely accumulation of contaminants such as Fe on the catalyst, which is very usual for 3d metal oxide catalysts during OER and greatly affects the activity.*

Response 3: We appreciate this valuable comment. First, the post-OER XRD results show the disappearance of CoP diffraction peaks, indicating that CoP may undergo structural transformation under OER process. This suggests that while CoP may serve initially as a conductive precursor or structural component, it likely reconstructs into other Co-based oxyhydroxide phases during anodic polarization, which are known to be active in alkaline OER. To better understand this transformation, post-OER Co 2p XPS analysis has been provided in the revised manuscript, which gives valuable insights

into the chemical state of Co after long-term electrolysis (**Fig. R5** or **Supplementary Fig. 27**). As the stability test time increases, the relative intensities of the Co $2p_{3/2}$ and Co $2p_{1/2}$ peaks increase, while those associated with low-valence $\text{Co}^{\delta+}$ species disappear, indicating progressive surface reconstruction into higher-valence Co species. In addition, low-valence Co can still be detected in the XPS spectra at a depth of 100 nm, but the peak intensity diminishes over time, suggesting an inward structural evolution wherein CoP is progressively transformed into hydroxide/oxide species.

Notably, electrochemical impedance spectroscopy measurements show that the charge transfer resistance remains nearly unchanged before and after the OER process (**Fig. R6** or **Supplementary Fig. 28**), indicating that the catalyst retains high electrical conductivity. This can be attributed to the relatively slow conversion of the CoP core into hydroxide/oxide species, allowing it to continuously serve as an efficient electron-conducting pathway. Additionally, the well-maintained one-dimensional nanowire structure also facilitates electron transport, further contributing to the overall conductivity.

The incorporation of Fe impurities from the electrolyte may significantly influence the structure and utilization of activity sites, representing a potential factor affecting its long-term stability. To evaluate this, we have conducted elemental distribution analysis using time-of-flight secondary ion mass spectrometry (TOF-SIMS) on the catalyst after 100-h electrolysis to determine whether Fe ions from the electrolyte will adsorb or incorporate into the catalyst (**Fig. R7**). As shown in the TOF-SIMS mapping images, the intensity of Fe distribution on the catalyst surface is extremely low, indicating that virtually no Fe adsorption occurs. Thus, the influence of Fe impurities on the catalytic activity and stability can be considered negligible.

Fig. R5 Depth- and catalytic reaction time-dependent XPS spectra of $\text{MnO}_2@Co\text{-Pi}@CoP$ in Co $2p$ region.

Fig. R6 Nyquist plots of $\text{MnO}_2@Co\text{-Pi}@CoP/NF$ after different reaction time.

Fig. R7 TOF-SIMS mapping of (a) total ion and (b) Fe fragments from $\text{MnO}_2@\text{Co-Pi}@\text{CoP}/\text{NF}$ after 100-h eASO.

Comment 4: *For educational purposes, I feel it is important that the authors correctly represent the "business case" for direct seawater electrolysis. The authors claim that pure water electrolyzers have to rely on fresh water. This is not the case. Seawater desalination is a well-established industrial process that can produce pure water for electrolysis, contributing only ~2% to the overall energy cost. The reason that direct seawater electrolysis could be of interest is that it reduces the footprint and CAPEX of the system, which might be interesting for e.g. offshore applications.*

Response 4: We sincerely appreciate the reviewer's insightful comment and agree that this is an important point that deserves clear articulation. In the revised manuscript (Introduction), we have corrected and clarified our rationale. Specifically, we acknowledge that seawater desalination is an industrially mature and economically feasible technology, and that the energy cost associated with desalinating seawater for electrolysis contributes ~2% to the total energy consumption of hydrogen production. Therefore, we agree that freshwater scarcity alone does not constitute a compelling justification for the development of direct seawater electrolysis. We have revised the manuscript to emphasize that the true value proposition of direct seawater electrolysis lies in system-level benefits. In particular, we highlight the potential to reduce the footprint, capital expenditure, and system complexity, which could be particularly advantageous for offshore or remote applications.

Comment 5: For the activity measurements, especially at these very high current densities, it is crucial to know what level of iR correction was applied. This is not reported with the data. However, the data in supp fig 5 looks over-corrected, particularly the red curve. This is also apparent from supp fig 6, where it can be seen that the apparent Tafel slope decreases at high current density, which is very unusual behavior. Later figures don't show this apparent iR correction and indeed look much more usual. In general, I am not sure whether a perfect iR correction is possible at very high current densities, given that the bubble behavior, local temperature, and pH gradients may drastically change the solution resistance. Therefore, for the discussion of the intrinsic overpotential of the catalyst, the authors should focus on the low-current density regime and only apply an iR correction here (and clearly state it). Other data should not be iR corrected.

Response 5: Thank you for your careful comment. At high current densities, bubble formation, local heating, and pH gradients can significantly affect the solution resistance, making accurate iR correction difficult and potentially misleading. Based on your suggestion, we have revised the manuscript to provide additional linear sweep voltammetry (LSV) curves with iR correction (85%) applied in the low-*j* region and uncorrected LSV curves to more accurately reflect the intrinsic activity of the catalyst (**Fig. R8a** or **Supplementary Fig. 6b**). At 10 mA cm⁻², the overpotentials are 266, 247, and 228 mV for CoP/NF, Co-Pi@CoP/NF, and MnO₂@Co-Pi@CoP/NF, respectively. According to the uncorrected LSV data (**Fig. R8b** or **Supplementary Fig. 6c**), their overpotentials follow the trends as CoP/NF (270 mV@10 mA cm⁻²) > Co-Pi@CoP/NF (250 mV@10 mA cm⁻²) > MnO₂@Co-Pi@CoP/NF (234 mV@10 mA cm⁻²). In addition, we have replaced the Tafel slopes in **Fig. R9** or **Supplementary Fig. 7** at low-*j* regions based on 85%-iR corrected LSV data. The results still clearly demonstrate that MnO₂@Co-Pi@CoP/NF exhibits superior OER kinetics compared to CoP/NF and Co-Pi@CoP/NF.

Fig. R8 (a) LSV curves with 85% iR correction at a low- j region. (b) LSV curves without any iR correction.

Fig. R9 Tafel plots of CoP/NF, Co-Pi@CoP/NF, and MnO₂@Co-Pi@CoP/NF at j of 10-100 and 100-200 regions.

Comment 6: Some description on how the TOF is calculated is necessary. Did the authors assume that any redox-active site is OER active?

Response 6: We sincerely thank the reviewer for their valuable comments. In the revised manuscript, we have added a detailed explanation of the TOF calculation method, including how the number of active sites was determined. As reported in previous works (*Angew. Chem. Int. Ed.* **60**, 23051–23067 (2021), *Adv. Mater.* **37**, 2410295 (2025)), estimating the number of active sites based on the integrated charge of redox peaks is a straightforward and effective approach. This is because only the truly catalytically active sites undergo redox transitions, thereby contributing to catalytic activity. Although our catalyst contains both Mn and Co elements, XPS analysis after different electrolysis durations revealed that the oxidation state of Mn

remained essentially unchanged, indicating that Mn may not participate in redox processes during OER. Therefore, we attribute the observed redox peaks primarily to the Co species. It is also well recognized that CoOOH typically serves as the main active phase for OER in alkaline seawater. Based on this understanding, we calculated the number of active sites by integrating the Co-related redox peak, which was then used in the TOF calculation.

Comment 7: *The activation energies are calculated from data obtained at very high overpotential, where mass/charge transport resistance plays a huge role. Therefore, the obtained values may reflect mass transport rather than the OER barrier. It would be better to do the calculation with low overpotential data.*

Response 7: We appreciate this comment. According to the suggestion, in the revised manuscript, we have provided the activation energies calculated with low overpotentials. As shown in Fig. R10 or Supplementary Fig. 11d, MnO₂@Co-Pi@CoP/NF shows a lower activation energy with 4.07 kJ mol⁻¹ at 1.6 V vs. RHE (without iR correction) compared with CoP/NF (11.1 kJ mol⁻¹) and Co-Pi@CoP/NF (11.01 kJ mol⁻¹), indicating its lower energy barrier to achieve OER.

Fig. R10 Arrhenius plots of the kinetic currents at 1.6 V vs. RHE.

Comment 8: *I think the authors may misinterpret their pH and cation dependence data. 3d metal oxides often become amorphous during the OER (the Co indeed seems*

to do that based on the XRD), after which cations can intercalate into the structure. This strongly modulates the OER performance of the material. When the pH is changed, the cation concentration is changed as well. Hence the difference between the different materials may primarily reflect the cation intercalation behavior.

Response 8: We sincerely thank the reviewer for the valuable comment. The pH and cation (TMA⁺) dependence experiments are two distinct approaches that serve a common purpose—to elucidate the reaction mechanism of the OER. The pH-dependence experiment is designed to investigate the proton–electron coupling behavior of the catalyst, while the TMA⁺-dependence experiment aims to quench oxygen intermediates and thereby assess whether the OER follows the lattice oxygen mechanism. The following provides a detailed explanation:

1. After carefully reviewing the literature (*Nat. Commun.* **14**, 1873 (2023), *Nat. Chem.* **9**, 457–465 (2017), *Nat. Energy* **4**, 329–338 (2019), *Nat. Commun.* **12**, 3992 (2021), *ACS Energy Lett.* **3**, 2884–2890 (2018)), we find that the pH dependence of OER activity is generally attributed to the proton–electron transfer mechanism rather than to the cation intercalation of amorphous oxyhydroxides. Specifically, under alkaline conditions, the OER involves multiple proton–charge-transfer steps, typically deprotonation of OH_(ads) or OOH_(ads) to O–O intermediates. These transformations are tightly coupled with proton and electron transfer processes. Variations in pH directly influence the formation energy and transition potentials of these intermediates, thus modulating the overpotential and kinetic rate constants. For instance, Surendranath et al. (*J. Am. Chem. Soc.* **132**, 16501–16509 (2010)) demonstrate that Co-Pi shows no pH-dependence of OER kinetics on the RHE scale, where proton-acceptor phosphate species governs the kinetics of surface deprotonation to enable high OER activity. In addition, amorphous transition metal oxyhydroxides show that the OER activity follows the same electrochemical principles with respect to deprotonation (*J. Am. Chem. Soc.* **137**, 15112–15121 (2015), *ACS Energy Lett.* **3**, 2884–2890 (2018)), leading to the formation of a negatively charged oxygenated intermediate (active oxygen) participating in the OER, rather than structural changes associated with

amorphization. Therefore, we believe that the pH dependence of OER activity should be attributed to the proton–electron transfer process rather than to the cation intercalation behavior.

2. TMA⁺ can selectively interact with electrophilic oxygen species on the catalyst surface without participating in proton–electron transfer processes. If lattice oxygen is involved in O–O bond formation, these reactive oxygen intermediates (e.g., *O or *OOH) can be effectively quenched by TMA⁺, leading to a noticeable suppression in OER activity. Therefore, a significant decrease in j with the addition of TMAOH provides strong evidence supporting the involvement of lattice oxygen oxidation during OER. (*Angew. Chem. Int. Ed.* **56**, 8652–8656 (2017), *Nat. Energy* **4**, 329–338 (2019), *Nat. Commun.* **12**, 3992 (2021)). It would be worth noting that the TMA⁺ experiment is independent of the pH-dependent experiment, and serves solely as a chemical probe for detecting oxygen intermediates associated with lattice oxygen participation.

Comment 9: Observing rare events like surface chlorination in AIMD is not likely. Therefore, I don't think the AIMD simulations carry much meaning for explaining the performance of the catalyst.

Response 9: We appreciate this comment. We truly appreciate your question. Currently, AIMD simulation with explicit solvent model has been widely employed to reveal the behavior of specific molecules and ions (e.g., O₂, CO₂, K⁺) during electrochemical process, and most results are consistent well with the experimental phenomena (*Nat. Catal.* **7**, 1000–1009 (2024), *Nat. Catal.* **7**, 1120–1129 (2024), *Nat. Commun.* **15**, 10877 (2024), *ACS Catal.* **14**, 3596–3605 (2024), *Nat. Commun.* **14**, 6936 (2023), *J. Am. Chem. Soc.* **145**, 25352–25356 (2023)). In our AIMD simulations, a long-term equilibrium period has been performed to ensure the rationality of our results, which agree well with our experiment. Besides, considering the abundance of *O intermediate on PO₄@MnO₂ (100) and γ-MnO₂ (100), it is likely to observe the formation of *OCl if PO₄@MnO₂ (100) and γ-MnO₂ (100) exhibit strong adsorption to Cl⁻.

Comment 10: *Are the durations reported in Fig. 3g the testing times, or the actual life spans of the catalysts? Of course, other materials may have also lasted for 3000 h, but were just not tested that long. This creates an unfair comparison. A better figure of merit would be the overpotential increase per hour.*

Response 10: We appreciate this comment. Fig. 3g shows the testing time of MnO₂@Co-Pi@CoP/NF and other reported catalysts. To create a fair comparison, in the revised manuscript, we included the decay voltage (D_V) values reported in the literature. Among them, only RuMoNi/NF (*Nat. Commun.* **14**, 3607 (2023)) and Os-Ni₄Mo/MoO₂/NF (*Adv. Mater.* **36**, 2408982 (2024)) provide D_V values of 0.64 and 0.37 $\mu\text{V h}^{-1}$, respectively, calculated using the formula $D_V = (V_1 - V_2)/t$, where V_1 and V_2 are averages of voltages from the first and last 10% operation time. However, the calculated V_1 and V_2 of MnO₂@Co-Pi@CoP/NF at 1 A cm⁻² are 2.13 and 2.04 V vs. RHE, respectively. This corresponds to a negligible voltage decay, indicating that MnO₂@Co-Pi@CoP/NF exhibits virtually no performance degradation for 3000-h electrolysis.

Comment 11: *In the Mn 2p analysis in Fig. 5b, the authors focus on the energy difference between the 2p_{3/2} and 2p_{1/2} to analyse the chemical state of the Mn. However, this spin-orbit splitting is essentially independent of the oxidation state (in contrast to the splitting of the Mn 3s, which is not due to spin-orbit coupling). It would make more sense if the authors look at the position of the Mn 2p_{3/2} peak.*

Response 11: Thank you for your valuable comment. To better understand the oxidation state evolution of Mn during eASO, XPS spectra of Mn 2p_{3/2} were collected at both the surface and approximately 100-nm depth after 0, 5, 20, and 100 h of operation. At the surface, the Mn 2p_{3/2} peak shows a gradual shift from 641.75 eV (initial) to 642.35 eV (after 100 h), reflecting a steady increase in the Mn oxidation state due to long-term anodic polarization. This positive shift toward higher binding energy is consistent with the formation of more oxidized Mn species (e.g., Mn⁴⁺), commonly observed under sustained eASO. In contrast, the Mn 2p_{3/2} peak at 100-nm depth remains relatively unchanged, exhibiting only a slight shift from 640.85 to 641.45 eV over 100 h. These observations indicate that even after 100 h of electrolysis,

the Mn remains predominantly in the +4 oxidation state, consistent with the preservation of the MnO₂ phase throughout extended eASO. A detailed discussion on the Mn 2p_{3/2} peak positions at both the surface and 100-nm depth, and their implications for the Mn oxidation state after long-term eASO, has been included in the revised manuscript.

Comment 12: Fig 1a-c are not very informative. Fig 1d is misleading, as it suggests that electrolysis in pure water + KOH has only been tested up to a few hundred hours. In fact this electrolyte has been used/tested in industrial alkaline electrolyzers over their full lifespan of ~8 years.

Response 12: Thank you for your insightful comments. **Fig. 1a–c** are primarily intended to convey the central design concept of this work, aiming to help readers clearly understand the core message of our study. We have added more explanatory details in the revised manuscript to clarify this intention. We sincerely apologize for the confusion caused by the current presentation of **Fig. 1d**. It was not our intention to suggest that electrolysis in pure water + KOH has only been evaluated over a few hundred hours. We fully recognize that alkaline electrolyzers using this electrolyte system have been widely adopted in industrial settings and have demonstrated stable operation over lifespans of several years. Our intention in Fig. 1d is not to suggest that the long-term durability of alkaline freshwater electrolysis systems is inherently limited to a few hundred hours. Rather, the examples highlighted in alkaline freshwater environments are primarily aimed at demonstrating catalyst design strategies that enhance mechanical stability—such as mitigating gas bubble adhesion and improving structural robustness—not chemical durability. These studies typically conducted tests over several hundred hours to validate the effectiveness of their structural optimization approaches, rather than to assess the full lifespan of the electrolyte or system. To avoid this misunderstanding, we will revise the figure and its caption in the revised manuscript, and we will clarify in the main text that the figure is meant to highlight the design motivations of recent works on catalyst surface/interface engineering for bubble management, rather than to imply limitations in electrolyte

stability. We greatly appreciate your feedback, which has helped us identify and correct this potential source of misinterpretation.

Comment 13: *Fig. 2 The images in b are recorded at a very different angle than c, making it difficult to assess if the morphology really changed. Arguably, the bottom image of b also show tip-connected arrays. Furthermore, for STEM-EDX analysis in f is would be good to add an overlay image of all the elements to highlight the core-shell structure.*

Response 13: We appreciate this comment. We appreciate the reviewer's valuable comments. To more intuitively and clearly illustrate the morphological differences, we selected SEM images taken at similar angles for direct comparison. As shown in **Fig. R11a** or **Fig. 2b**, the CoP nanowires predominantly exhibit a two-dimensional arrangement, with signs of intertwining and collapse. In contrast, **Fig. R11b** or **Fig. 2c** clearly reveals an ordered array of nanocage structures, which is beneficial for bubble release. In addition, we have included the overlaid STEM-EDX mapping images (**Fig. R12** or **Fig. 2f**), which show that Mn is mainly distributed at the edges of the nanowires (shell), while Co is more concentrated in the core region, confirming a well-defined core-shell structure.

Fig. R11 SEM images of (a) CoP/NF and (b) MnO₂@Co-Pi@CoP/NF. Scale bar: 1 μ m (upper) and 500 nm (lower).

Fig. R12 Scanning TEM and the mapping images of MnO₂@Co-Pi@CoP. Scale bar: 200 nm.

Comment 14: *It is unclear what is shown in fig. 3f AIMD.*

Response 14: We appreciate this comment. The AIMD results presented in **Fig. 3f** show the distance between Cl⁻ and the surfaces of different catalysts over time, providing insight into the chloride-repelling capability of various exposed facets. As shown in **Fig. 3f**, the distance between Cl⁻ and the MnO₂ surface remains consistently large throughout the simulation, indicating that Cl⁻ cannot adsorb onto MnO₂, which demonstrates its strong Cl⁻-repelling ability. In contrast, for CoP and Co-Pi, the Cl⁻ remains close to the surface (at around 2 Å), suggesting a lack of effective repulsion and a tendency for Cl⁻ to adsorb onto the catalyst surface. In the case of CoOOH, its relatively strong Lewis acidity promotes OH⁻ adsorption, which contributes to a certain degree of Cl⁻ repulsion. However, this repelling effect is not as stable or pronounced as that observed for MnO₂.

Comment 15: *Supp. fig. 1: please note what materials the PDFs belong to.*

Response 15: We appreciate this comment. According to the suggestion, in the revised manuscript, we have updated the corresponding information of the PDFs in **Supplementary Fig. 1**.

Comment 16: *Supp. fig. 9d: should the unit be V instead of mV?*

Response 16: We appreciate this comment. According to the suggestion, we have corrected the errors in **Supplementary Fig. 10d** (updated number of **Supplementary Fig. 9d**).

Point-by-point response to the reviewers #2

Reviewer #2 (Remarks to the Author): While we previously expressed concern about the very limited value of "lifetime rates" in the OER catalyst field, the authors continue to assert that reporting long-term stability remains meaningful. If this is the case, Figure 1 in the main text should comprehensively include representative and high-performance literature data for a fair and informative comparison. Although some of this information is provided in Supplementary Table 1, the current form of Figure 1 remains unrevised and thus misleading. To improve transparency and ensure balanced interpretation, key prior results should be accurately and visibly plotted in a "current density vs. lifetime" format in Figure 1, without any intentional omission (*Nat. Commun.* 15, 2481 (2024) for 3000 h at 0.5 A/cm²; *Nat. Sustain.* 7, 158–167 (2024)) for a 2800-h duration at 1.25 A/cm²; *Adv. Mater.* 36, 2411302 (2024) for 10000 h at 0.4 A/cm²; *Adv. Mater.* 37, 2415421 (2025) for 2500 h at 1 A/cm²; and *Nature* 639, 360–367 (2025) for 10,000-h stability in intermittent seawater electrolysis).

Response: We appreciate this comment and do understand your concerns. Fig. 1g is intended to illustrate the different mechanisms by which the stability of recently reported catalysts has been improved, rather than to provide a direct performance comparison. In response to the reviewer's request, we have added the relevant references in Fig. 1 and/or Supplementary Table 1.